# ⌒ CHEF: A COMPREHENSIVE EVALUATION FRAME- WORK FOR STANDARDIZED ASSESSMENT OF MULTI- MODAL LARGE LANGUAGE MODELS

## ABSTRACT

Multimodal Large Language Models (MLLMs) have shown impressive abilities in interacting with visual content with myriad potential downstream tasks. However, even though a list of benchmarks has been proposed, the capabilities and limitations of MLLMs are still not comprehensively understood, due to a lack of a standardized and holistic evaluation framework. To this end, we present the first *Comprehensive Evaluation Framework* (ChEF) that can holistically profile each MLLM and fairly compare different MLLMs. First, we structure ChEF as four modular components, *i.e.*, *Scenario* as scalable multimodal datasets, *Instruction* as flexible instruction retrieving formulae, *Inferencer* as reliable question-answering strategies, and *Metric* as indicative task-specific score functions. Based on them, ChEF facilitates versatile evaluations in a standardized framework, and new evaluations can be built by designing new *Recipes* (systematic selection of these four components). Notably, current MLLM benchmarks can be readily summarized as recipes of ChEF. Second, we introduce 6 new recipes to quantify competent MLLMs' desired capabilities (or called desiderata, *i.e.*, calibration, in-context learning, instruction following, language performance, hallucination, and robustness) as reliable agents that can perform real-world multimodal interactions. Third, we conduct a large-scale evaluation of 9 prominent MLLMs on 9 scenarios and 6 desiderata. Our evaluation summarized over 20 valuable observations concerning the generalizability of MLLMs across various scenarios and the composite capability of MLLMs required for multimodal interactions. We will publicly release all the detailed implementations for further analysis, as well as an easy-to-use modular toolkit for the integration of new recipes and models, so that ChEF can be a growing evaluation framework for the MLLM community.

## 1 INTRODUCTION

By applying the powerful Large Language Models (LLMs) (OpenAI, 2023; Chiang et al., 2023; Touvron et al., 2023) as a universal task interface, recent works on Multimodal Large Language Models (MLLMs) (Liu et al., 2023a; Zhu et al., 2023; Dai et al., 2023) have shown impressive abilities to interact with visual contents through question-answering dialogues and are expected to address more complex multimodal tasks that can harness LLMs' generalization ability to myriad downstream scenarios. Yet the capabilities and limitations of MLLMs are still not well understood, and we observe a lack of a standardized framework that can comprehensively evaluate different MLLMs. Recent benchmarks often focus on building a multimodal evaluation dataset for MLLMs (Li et al., 2023b; Liu et al., 2023c; Fu et al., 2023) or only evaluate one or a few factors of MLLMs (Shao et al., 2023; Li et al., 2023d; Yu et al., 2023; Bitton et al., 2023), or attempt to establish a framework but lack scalability and have limits in their comprehensiveness (Yin et al., 2023; Xu et al., 2023) [1]. This makes a thorough assessment of each model and reliable comparisons among various models challenging.

To address these issues, we believe that a comprehensive evaluation framework, which is specially designed for MLLMs, should encompass scalable datasets about multimodal tasks that can be han-

---

[1]More related works are provided in Supplementary Materials (Section A).

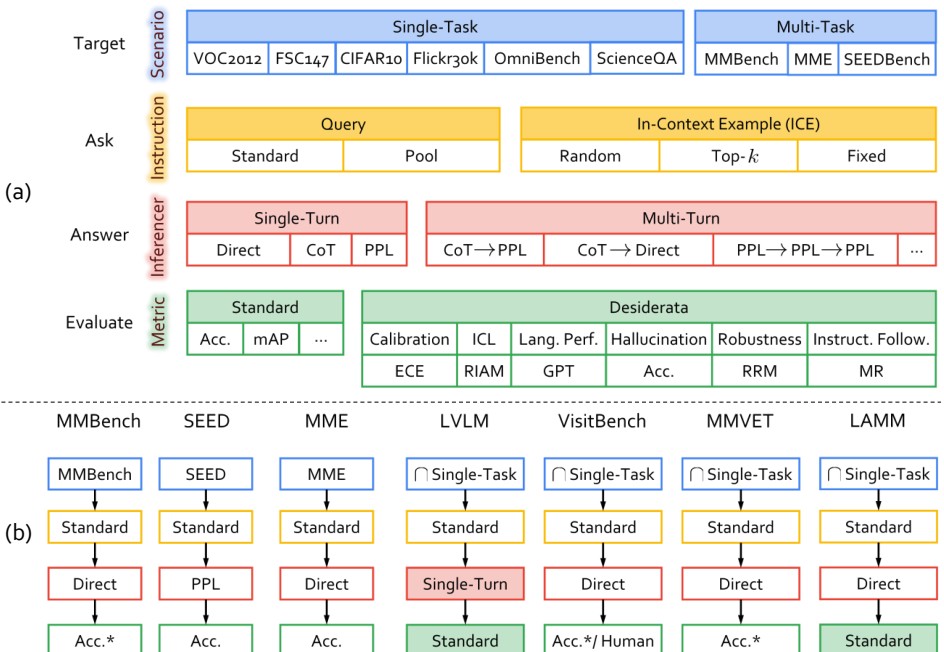

Figure 1: (a) ChEF Overview. (b) Current MLLM benchmarks can be readily absorbed into ChEF. *Acc.* is the accuracy. *Acc.\** is the accuracy from GPT-based metric. ∩ means overlap with ChEF. *ICL, Lang. Perf., Instruct. Follow.* are shorts for in-context learning, language performance, and instruction following, respectively.

dled by MLLMs. For each model, we should evaluate the performance in a broad set of perspectives (*i.e.* capabilities more than multimodal perception and reasoning, such as robustness, in-context learning, and *etc.*) that are vital to profile the intrinsic properties of MLLMs, especially as the agents that can perform real-world multimodal interaction. Moreover, meaningful comparisons among MLLMs require standardization in the evaluation process so that each model can be conveniently adapted. To this end, as shown in Figure 1(a), we present ChEF, a Comprehensive Evaluation Framework for reliable and indicative assessment of MLLMs, which is highly scalable and can be flexibly modified to adapt to the evaluation of any new model or task. It is modularly designed with four components, *i.e.*, *Scenario*, *Instruction*, *Inferencer*, and *Metric*.

**(1) Scenarios** are a set of datasets concerning representative multimodal tasks that are suitable for MLLMs. *Scenarios* are scalable by design, allowing the inclusion of any related dataset if necessary. We have included several prominent single-task datasets, such as CIFAR-10 (Krizhevsky & Hinton, 2009) for image classification, VOC2012 (Everingham et al., 2012) for object detection, ScienceQA (Lu et al., 2022) for multimodal question-answering. Recent multi-task benchmark datasets proposed for evaluating MLLMs, such as MMBench (Fu et al., 2023) and SEEDBench (Li et al., 2023b), are also accessible as *Scenarios*.

**(2) Instruction** focuses on how to pose questions and set instruction examples to the MLLMs. We integrate various standard queries and query pools adaptive to each MLLM, and multimodal in-context example (`ICE`) retrieving strategies for in-context learning (ICL) (Wu et al., 2023; Brown et al., 2020). Both are tailored to specific *Scenarios*. To the best of our knowledge, we are the first to incorporate ICL into the evaluation framework. The design of *Instruction* makes it flexible to evaluate diverse *Scenarios* within the same framework.

**(3) Inferencer** pertains to how an MLLM answers questions. In a single-turn question-answering (QA), in addition to the standard textual outputs (`Direct`) that may be hard to compare with the ground-truth answers, we can employ the Perplexity (`PPL`) (Klein et al., 2017) to select the most probable candidate answers, or Chain-of-Thought (`CoT`) (Zhang et al., 2023) prompting to increase the reliability of the prediction. The *Inferencer* also allows `Multi-Turn`, in which `PPL`, `CoT`, and `Direct` outputs can be applied in turns, and makes the evaluation result reliable.

**(4) Metrics** are a set of score functions designed to evaluate the performance of each MLLM. For example, we include task-specific metrics such as accuracy for classification or multi-choice QA,

mAP for detection, BLEU for captioning, and *etc.* More metrics can be included when evaluating the MLLMs from new perspectives, such as Expected Calibration Error (ECE) (Naeini et al., 2015) if we would like to know how the model is aware of its uncertainty in prediction, GPT-based metric (Chiang & Lee, 2023) if we would like the outputs to be readable as natural language. The inclusion of appropriate and newly defined metrics ensures that the evaluation results are more indicative.

With a systematic selection of *Scenarios*, *Instructions*, *Inferencers*, and *Metrics*, ChEF facilitates versatile evaluations in a standardized framework. Users can easily build new evaluations according to new *Recipes* (*i.e.* specific choices of the four components). For example, current MLLM benchmarks (Fu et al., 2023; Li et al., 2023b; Liu et al., 2023c; Bitton et al., 2023; Yu et al., 2023; Xu et al., 2023; Yin et al., 2023) can be summarized as different *Recipes*, as shown in Figure 1(b), and thus can be readily absorbed into ChEF. We will extensively discuss the design principles in Section 2.1. Moreover, we view ChEF as a growing framework, where each component can be evolved according to the emerging techniques or applications. We will continuously update the ChEF framework with a wider range of accessible models and evaluation tasks.

Based on ChEF, it becomes rather convenient to set up new evaluations to quantify the desired capabilities (or called **desiderata**) that a competent MLLM model should possess, as a reliable agent that can perform real-world multimodal interactions. These desiderata include:

- **Calibration**: Does MLLM express accurate uncertainty and confidence?
- **In-context Learning**: Does MLLM learn from instruction examples?
- **Instruction Following**: Does MLLM adhere to instructions?
- **Language Performance**: Does MLLM describe visual content in readable language?
- **Hallucination**: Does MLLM avoid mentioning objects that do not exist in the images?
- **Robustness**: Is MLLM robust to corruptions in the multimodal inputs?

Each desideratum is evaluated by constructing the evaluation pipeline from a ChEF *Recipe*. We will introduce the *Recipes* for the desiderata in Section 2.3.

Overall, we comprehensively evaluated 9 MLLMs across 9 *Scenarios* and 6 desiderata. Our evaluation yields the following 3 key findings:

**(1)** Recent MLLMs cannot perform well across all *Scenarios*. There is a significant tug-of-war issue (Hadsell et al., 2020) between different tasks. There are also several critical tasks that can not be addressed by recent MLLMs.

**(2)** Recent MLLMs are struggling with in-context learning, instruction following, and robustness, thus they may fall short of real-world multimodal interactions.

**(3)** There is a strong correlation between the desiderata and visual performance. Evaluating the desiderata reveals the intrinsic property on *Scenarios* that used to evaluate a composite performance.

## 2 CHEF: A COMPREHENSIVE EVALUATION FRAMEWORK

In this section, we first list the design principles of ChEF in Section 2.1, and then depict how to conduct an evaluation process based on a *Recipe* of selecting the four modules in ChEF (Section 2.2). Furthermore, we introduce the *Recipes* of six desired capabilities (or called desiderata) that a competent MLLM should have, as shown in Section 2.3.

### 2.1 DESIGN PRINCIPLES

ChEF is a comprehensive evaluation framework aiming at providing a fair and holistic assessment of MLLMs' performance across diverse multimodal tasks. To accomplish this objective, our design principles encompass the following key aspects:

**(1) Modular.** We decouple the evaluation framework into four modular components [2]: *Scenario*, *Instruction*, *Inferencer*, and *Metric*, so as to enable fast modification of each component and ensure consistent evaluation results across different benchmark datasets.

**(2) Scalable.** We implement easy-to-use interfaces to streamline the integration of new *Scenarios* into the framework and have included almost all recent benchmark datasets into the *Scenario*.

---

[2]Details of these four components are provided in Supplementary Materials (Section B).

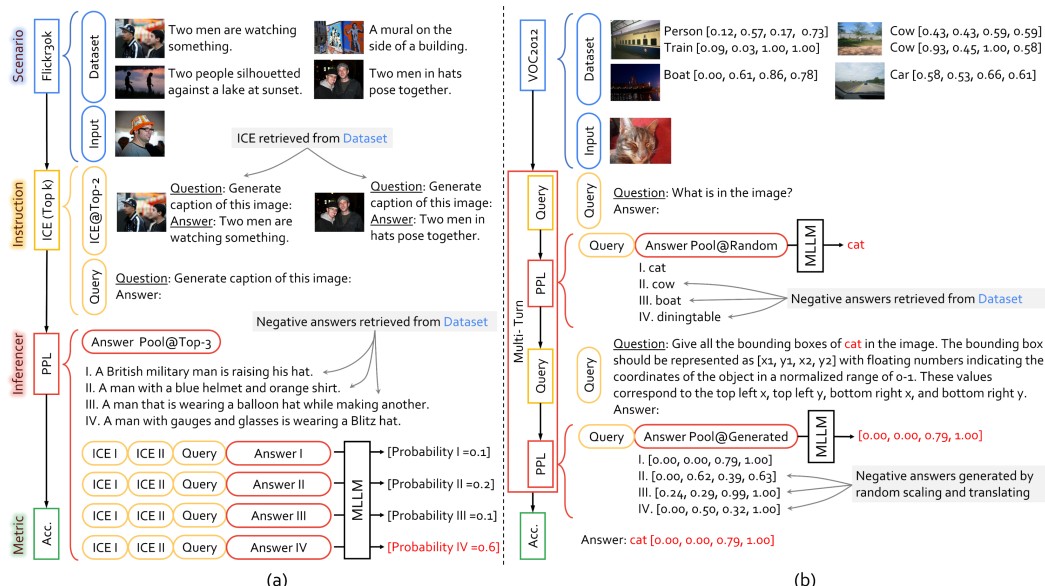

Figure 2: **Two examples of Recipes in ChEF.** A *Recipe* consists of {*Scenario*, *Instruction*, *Inferencer*, *Metric*}. The *Recipe* of (a) is {Flickr30k, `ICE`, `PPL`, Accuracy}, while (b) is {VOC2012, `Query`, `Multi-Turn`, Accuracy}.

**(3) Flexible.** We design ChEF to accommodate the varying input formats supported by different MLLMs, including `Queries` and in-context learning examples (`ICE`). Based on these *Instructions*, MLLMs can generate outputs that are suitable for specific *Scenarios*.

**(4) Reliable.** We include three more reliable *Inferencers*, such as `CoT` and `PPL`, as well as their multi-round combination (`Multi-Turn`), in addition to standard free-form outputs (`Direct`). These *Inferencers* make the evaluation more reliable, and better tailored to reflect the precise perception or reasoning abilities that the *Scenarios* tend to assess.

**(5) Indicative.** We utilize a list of task-specific metrics ranging from metrics for vision tasks to the GPT-based metric for language proficiency. Each MLLM's textual outputs are adapted to these metrics, so as to indicatively measure that the MLLMs can actually perform the target tasks.

## 2.2 EXEMPLAR RECIPES AND THEIR EVALUATION PROCESSES

For an illustration of how each component functions and the overall evaluation is processed, we provide two examples of *Recipes* in Figure 2.

**(1) Image Captioning on Flicker30k.** In Figure 2(a), the *Scenario* is Flickr30k and the task is image captioning. The *Instruction* does not only include the standard query "Generate caption of this image", but also Top-$k$ `ICE` to guide the generation of captions. These examples are retrieved according to image similarity. The *Inferencer* applies single-round `PPL` to measure how each of the four answers (as the answer pool) is consistent with the input image, in the form of probability. The negative answers are retrieved based on text similarity. Using `PPL` instead of free-form outputs constrains the scope of the captions and thus can be measured more reliably. Finally, to be compatible with `PPL`, the *Metric* applies accuracy to determine the correctness of the prediction.

**(2) Object Detection on VOC2012.** Object detection is another typical vision task. In Figure 2(b), we apply VOC2012 as the *Scenario*. The *Instruction* has no `ICE`, but just a standard query. The *Inferencer* is `PPL` that is conducted in two rounds. In the first round, ask the MLLMs "What is in the image?", and in the second round, ask the MLLMs the bounding box of the predicated object. Note that the answer pools of the bounding boxes are generated by random scaling and translating the ground-truth bounding boxes. The *Metric* is accuracy as we transform the detection task into a multi-choice question-answering paradigm.

## 2.3 DESIDERATA

As shown in Figure 3, we implement six more evaluations based on the desiderata that a competent MLLM model should have, *i.e.*, calibration, in-context learning, instruction following, language

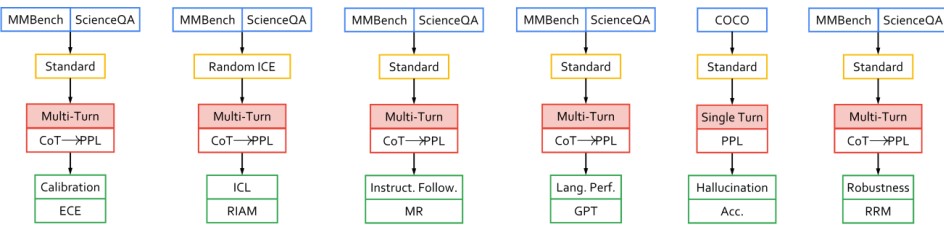

Figure 3: **Recipes for evaluating six dimensions of desiderata.** 1) All six dimensions are assessed on MMBench and ScienceQA, except for Hallucination, which is evaluated solely on MSCOCO; 2) All use standard query as *Instruction*, except ICL uses random `ICE`; 3) All employ `Multi-Turn` from `CoT` to `PPL` as *Inferencer*, except Hallucination with a single `PPL`; 4) The *Metric* for each dimension is specifically designed for the respective evaluation.

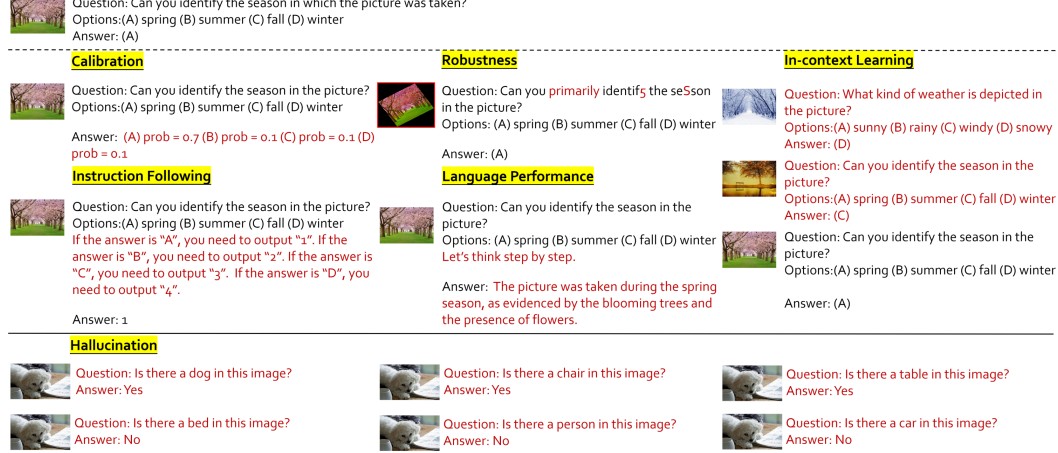

Figure 4: **The exemplar of desiderata.** The distinguished design of each desideratum is marked in red. For calibration evaluation, the prediction confidence is calculated to determine the gap between confidence and accuracy. Instruction following is evaluated through verbalizer manipulation. In-context learning is evaluated by providing `ICE` in the *instruction*. Robustness is assessed by introducing noise to both the image and text inputs. Language performance is evaluated by instructing the model to generate chain-of-thought content. Hallucination is solely evaluated on MSCOCO, and evaluated by querying whether a specific object is present in the image.

performance, robustness, and hallucination. Each dimension is assessed using a specially designed *Recipe*. To fulfill consistent evaluations among different dimensions of the desiderata, the *Scenarios* are almost MMBench (Liu et al., 2023c) and ScienceQA (Lu et al., 2022), except that hallucination is evaluated on MSCOCO (Lin et al., 2014). The *Inferencers* share a similar strategy. Hallucination applies `PPL` in a single round, while the rest desiderata use the same `Multi-Turn` that is composed of `CoT` and `PPL`, to increase the reliability of the prediction. In the following part, we introduce the rest components in each *Recipe*.

**(1) Calibration.** It evaluates how the uncertainty about each MLLM's prediction is aligned with its accuracy, as highlighted by HELM (Liang et al., 2022). As shown in Figure 4, its *instruction* is a standard query. Moreover, calibration is measured using the Expected Calibration Error (ECE) (Naeini et al., 2015; Guo et al., 2017), which calculates the difference between the model's predicted probability and the fraction of times the model is correct.

**(2) In-context Learning.** It evaluates the crucial in-context learning (ICL) ability of an MLLM. To evaluate this desideratum, the *Instruction* is set to include randomly retrieved in-context examples (`ICE`). Note that `ICE` can include images. To assess the ICL ability, we introduce the Relative ICL Accuracy for Multi-Choice QA (RIAM), which measures the relative accuracy improvement beyond random guessing, written as

$$\text{RIAM} = (\text{acc}_{\text{ICL}} - \text{acc}_{\text{0-shot}}) / (\text{acc}_{\text{0-shot}} - \text{acc}_{\text{rand}}), \tag{1}$$

where $\text{acc}_{\text{ICL}}$ denotes the average accuracy among the results based on the instructions with different shots of in-context examples. $\text{acc}_{\text{0-shot}}$ means zero-shot prediction without `ICE`. $\text{acc}_{\text{rand}}$ means the accuracy by random guessing.

**(3) Instruction Following.** It evaluates how exactly the MLLM relies on the given instructions. The *Instruction* is set as standard query, which is retrieved from the three categories of instructions as the way used in verbalizer manipulation, *i.e.*, *natural*, *neutral*, and *unnatural* (Li et al., 2023c). The *Metric* applied here is the Match Ratio (MR), which calculates the percentage of textual outputs that are matched with the outputs indicated by the verbalizer instructions.

**(4) Language Performance.** It evaluates the quality of the generated sentences. Since the applied *Inferencer* does not generate free-form output, we evaluate the language performance of the outputs corresponding to the chain-of-thought. Knowing that GPT-based metrics have shown to be well correlated with human evaluation (Zheng et al., 2023; Liu et al., 2023b; Wang et al., 2023), we use GPT-4 to evaluate the language performance of the `CoT` outputs based on the ground-truth sentences (*i.e.* questions and answers) in the question-answering tasks. Moreover, we choose the average results of multiple rounds of evaluations to eliminate the flickering of the GPT-based evaluations.

**(5) Robustness.** It measures how robust an MLLM is to corruptions in the multimodal inputs. The image corruptions include noise, blur, weather, digital (Hendrycks & Dietterich, 2019) and common data augmentation techniques. The textual corruptions include sentence-, word- and character-level corruptions (Chen et al., 2023b), as well as switching choices for multi-choice question-answering.

The *Metric* in this desideratum is Relative Robustness for Multi-Choice (RRM), written as

$$\text{RRM} = (\text{acc}_{\text{crp}} - \text{acc}_{\text{rand}})/(\text{acc} - \text{acc}_{\text{rand}}), \tag{2}$$

where $\text{acc}_{\text{crp}}$ denotes the accuracy after corruption, acc is the accuracy before corruption. $\text{acc}_{\text{rand}}$ means the accuracy by random guessing.

**(6) Hallucination.** It evaluates how an MLLM avoids mentioning visual objects that do not exist in the images. The *Scenario* is MSCOCO. We follow the Polling-based Object Probing Evaluation (POPE) (Li et al., 2023d) in this desideratum. It transforms hallucination evaluation into a set of binary classification tasks. Essentially, the MLLMs are posed Yes-or-No questions about the existence of some particular objects in the images, such as "Is there a car in the image?" Notably, `PPL` is applied to as a more reliable *Inferencer*. The *Metric* applied here is accuracy.

## 3 EXPERIMENTS

### 3.1 EVALUATION SETUP

A wide range of recently introduced MLLMs are evaluated in ChEF, including LLaVA (Liu et al., 2023a), LAMM (Yin et al., 2023), MiniGPT4 (Zhu et al., 2023), mPLUG-Owl (mPLUG) (Ye et al., 2023), Otter (Li et al., 2023a), InstructBLIP (Dai et al., 2023), LLaMA-Adapter-v2 (LAv2) (Gao et al., 2023), as well as models specifically designed for grounding tasks, such as Shikra (Chen et al., 2023a) and Kosmos-2 (Peng et al., 2023). These MLLMs are evaluated across various single-task *Scenarios*, including CIFAR-10 (CIFAR) (Krizhevsky & Hinton, 2009) for classification, Omnibenchmark (Omni) (Zhang et al., 2022) for fine-grained classification, VOC2012 (VOC) (Everingham et al., 2012) for object detection, FSC147 (FSC) (Ranjan et al., 2021) for object counting, Flickr30k (Flickr) (Young et al., 2014) for image captioning and ScienceQA (SQA) (Lu et al., 2022) for multimodal question-answering. We also evaluate the MLLMs on several multi-task datasets including MME (Fu et al., 2023), MMbench (MM) (Liu et al., 2023c) [3], and Seedbench (SEED) (Li et al., 2023b).

### 3.2 STANDARD PERFORMANCE OF VISUAL ABILITY

For each *Scenario*, we conduct various experiments with diverse *Recipes*, from which, the *Recipe* behaving most reliably (*i.e.* stable to *Instruction* variations) is selected as the default setting [4] to

---

[3]MMBench provides two evaluation settings (*i.e.*, VanillaEval and CircularEval). VanillaEval is adopted in the default *Recipe*.

[4]The default *Recipe* is also demonstrated to display and approach the best performance of each MLLM, as shown in Figure 6(a-b).

Table 1: **Visual performance of MLLMs on different Scenarios.** In SQA and MM, as options {A, B, C, D} are explicitly provided in the questions, models are required to output their answers in the form of options. Similarly, MME also requires models to provide "yes" or "no" outputs. These *Scenarios* can be considered as a discriminative (discrim.) question type. Conversely, the other *Scenarios* are characterized by generative (gen.) types, as they require responses without predefined options in questions. The abbreviations for *Scenarios* and MLLMs are defined in section 3.1. For Omnibenchmark (Omni$^\dagger$), weighted accuracy is employed, which entails a weighted accuracy calculation based on the granularity of classification. The entries that are both bold and underlined indicate the best performance.

| Scenario | CIFAR | Flickr | VOC | Omni$^\dagger$ | FSC | SQA | MM | SEED | MME |
|---|---|---|---|---|---|---|---|---|---|
| **Question Type** | gen. | gen. | gen. | gen. | gen. | discrim. | discrim. | gen. | discrim. |
| **LLaVA** | **89.40** | 80.80 | 26.01 | 26.62 | 24.11 | 46.55 | 43.13 | 46.45 | 50.17 |
| **LAMM** | 80.70 | 72.50 | 29.58 | 22.54 | 19.33 | 52.75 | 44.47 | 47.03 | 55.82 |
| **MiniGPT-4** | 80.80 | 71.50 | 26.51 | 30.60 | 22.52 | 47.0 | 54.34 | 46.48 | 57.12 |
| **mPLUG** | 79.67 | 79.20 | 28.50 | 30.70 | 20.92 | 48.44 | 49.57 | 42.81 | 71.59 |
| **Otter** | 81.34 | 71.30 | 27.15 | 26.41 | 20.00 | 50.22 | 53.91 | 36.40 | 63.78 |
| **LAv2** | 70.17 | 79.50 | 31.60 | **32.00** | 21.26 | 54.34 | 57.06 | 35.41 | 69.90 |
| **InstructBLIP** | 84.27 | 79.40 | 27.65 | 30.75 | **25.04** | **55.18** | **65.73** | **50.81** | **72.0** |
| **Shikra** | 68.71 | **94.70** | **55.23** | 22.89 | 22.43 | 45.21 | 63.26 | 49.79 | 70.28 |
| **Kosmos-2** | 88.87 | 85.70 | 54.55 | 21.34 | 21.93 | 34.60 | 25.60 | 46.38 | 52.95 |
| **Random Choice** | 10.0 | 25.00 | 25.00 | 10.94 | 20.00 | 35.80 | 27.57 | 24.27 | 50.00 |

evaluate the visual performance of all MLLMs, as shown in Table 1. As the default *Recipes* incorporate `PPL`, which can be regarded as a multi-choice question-answering paradigm, we also provide the accuracy of random choice for each *Scenario*. There are some observations as follows:

**(1)** InstructBLIP attains superior performance across most *Scenarios*. It is worth noting that both Shikra and InstructBLIP showcase exceptional performance on the multi-task datasets, including MME, MMBench, and SEEDBench, while the performance of other models displays inconsistencies. The visual performance of these MLLMs exhibits strong trade-offs across different tasks.

**(2)** All the MLLMs struggle in the object counting task (*i.e.* FSC), primarily due to the complexities associated with the precise identification of numerous objects within an image.

**(3)** There is a capability gap between detection and other tasks. Shikra and Kosmos-2 demonstrate remarkable detection capabilities, owing to their specialized training on detection datasets. However, Kosmos-2 exhibits limited aptitude in other *Scenarios*, especially on MMBench and ScienceQA. Despite its ability to perform perception and reasoning tasks, Kosmos-2 struggles to comprehend the meaning of options {A, B, C, D} provided in the question, resulting in difficulty in aligning the answers to options. As a consequence, it exhibits lower performance on discriminative tasks.

The unified evaluation of these models on diverse *Scenarios* in the ChEF enables us to conduct a fair comparison, discerning the optimal architectures and methodologies for specific *Scenarios*.

## 3.3 RESULTS OF DESIDERATA

The scores of all the desiderata on MLLMs are shown in Figure 5 with the corresponding accuracy of MMBench which we consider as the most representative assessment of MLLMs' visual performance. The six dimensions of desiderata are deemed essential for an MLLM to function as an interactive AI agent, emphasizing human-like interactions. However, the poor performance on these dimensions shows that current MLLMs fall short of being an AI agent capable of interacting with humans.

**(1)** Most MLLMs exhibit good calibration, indicating their ability to accurately convey uncertainty. This is primarily due to the relatively low accuracy of these models and their lack of confidence in the responses, which results in such consistency.

**(2)** Most MLLMs achieve satisfactory language performance, except for Kosmos-2, which provides few reasoning processes in its chain-of-thought responses.

**(3)** InstructBLIP and Shikra surpass other models on hallucination and meanwhile achieve superior visual performance on MMBench, emphasizing the crucial role of hallucination.

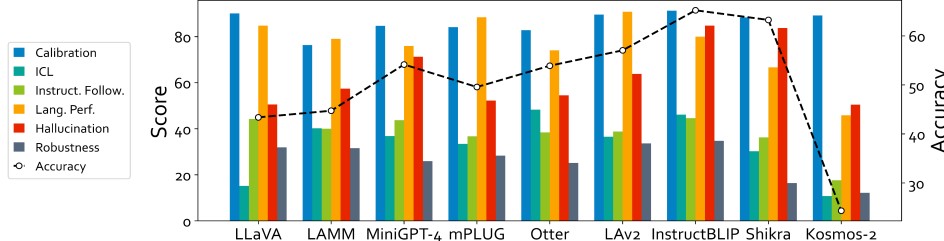

Figure 5: **Results of desiderata.** The dashline is the accuracy evaluated on MMBench. The score for each dimension is computed by normalizing the results from the specific metric to a range of 0-100. Calibration score is represented by 1-ECE. Instruction following score is the average MR across different verbalizer settings. In-context learning score is the average RIAM across various shot numbers. Language performance score is normalized from the results of the GPT-based metric. Robustness score is normalized from RMM and hallucination score directly represents accuracy.

**(4)** Most MLLMs exhibit poor performance in ICL. Notably, Otter, which is specifically trained on in-context instruction tuning data, though performs the best ICL among the 9 MLLMs, also struggles in ICL primarily due to its limited proficiency in visual tasks.

**(5)** Instruction following and robustness pose challenges for most MLLMs in effectively handling *Instructions* that deviate from their priors and their susceptibility to noisy multimodal inputs.

## 3.4 CHEF PROVIDES STABLE ASSESSMENT

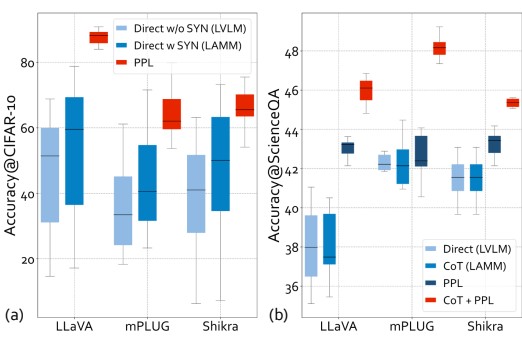

Figure 6: Results of various *Inferencers* across different queries on CIFAR10 and ScienceQA. Black lines within each boxplot represent the median. Boxplots display the accuracy distribution.

Due to the modular design of ChEF, it has the flexibility to employ different *Recipes* for evaluating the same *Scenario*. To get a reliable and fair evaluation, we conduct exhaustive experiments to identify the *Recipe* that behaves more stable on *Instruction* variations than previous approaches as the default setting.

Two examples, shown in Figure 6, are conducted on CIFAR10 and ScienceQA with distinct *Recipes* for three MLLMs. Figure 6(a) shows that utilizing `Direct` as *Inferencer* proposed in LAMM (Yin et al., 2023) (with the inclusion of synonyms judgment in the metric) and LVLM (Xu et al., 2023) (without synonyms) with different queries yields a large variance. Alternatively, employing the `PPL` can substantially mitigate these fluctuations with a much smaller variance, accompanied by a noteworthy gain in accuracy for all MLLMs. Similar observations can be also found in Figure 6(b). We further leverage `CoT`, which mandates the model to provide its reasoning process. Although the accuracy has a slight gain, it does not bolster the stability. Nevertheless, the optimal combination of accuracy and stability emerges when employing both the `CoT` and `PPL` in a `Multi-Turn Inferencer`.

Based on these interesting discoveries, we believe that ChEF, in conjunction with the meticulously derived and recommended *Recipes* for diverse *Scenarios*, can deliver a trustworthy and indicative assessment of MLLMs. We also conduct numerous experiments to carefully select appropriate *Recipes* for reliable evaluations across the six dimensions of desiderata [5].

## 3.5 CORRELATION BETWEEN VISUAL PERFORMANCE AND DESIDERATA

To investigate the relationship between visual performance and the desiderata, we display the Pearson correlation matrix in Figure 7(a).

---

[5]More evidence of reliability is provided in the Supplementary Materials (Section F).

**(1)** Calibration is an independent dimension, primarily assessing a model's proficiency in expressing uncertainty, without direct correlations to other dimensions.

**(2)** ICL demonstrates correlation with others, as their evaluations involve specific instructional aspects. MLLMs with enhanced ICL ability are better equipped to provide relevant responses to unseen cases.

**(3)** Instruction following demonstrates a significant correlation with language performance, robustness, and accuracy. As language performance assesses the content of an

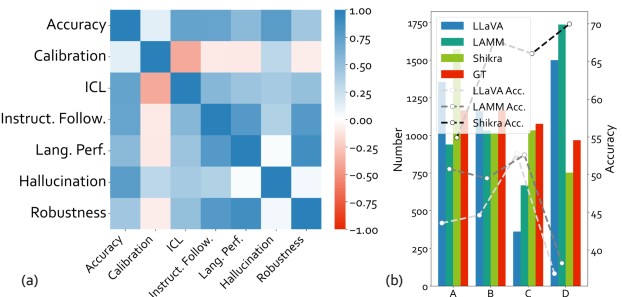

Figure 7: (a) Pearson correlation matrix of desiderata and accuracy on MMBench. Cooler colors indicate higher correlations. (b) Choice distribution with accuracy on MM-Bench. GT indicates the actual choice distribution.

MLLM's reasoning process, which is obtained through instructional guidance, MLLMs with stronger instruction following capabilities are more likely to adhere to the "step by step" instruction and generate a comprehensive reasoning process.

**(4)** Hallucination is strongly correlated with the performance on MMBench. The choice distribution of three models, as shown in Figure 7(b), reveals that LLaVA and LAMM prefer option D to C, while Shikra tends to favor option A over D. These MLLMs display lower accuracy on options they are inclined to answer and perform better on options that they resist. The distinct prior to options, which is caused by the hallucination issue, leads to poor performance.

It can be concluded that the evaluation of *Scenarios* that involve discriminative questions evaluates a composite performance, *i.e.*, visual performance, and additional dimensions of abilities, such as the comprehension of options. The evaluation of desiderata unveils intrinsic properties beyond visual performance.

## 4  CONCLUSION

In this work, we introduce ChEF, a comprehensive evaluation framework for holistically profiling and comparing MLLMs. ChEF's modular design (*i.e. Scenario*, *Instruction*, *Inferencer*, and *Metric*) enables versatile evaluations in a standardized framework. Based on ChEF, any evaluation, including current MLLM benchmarks, can be summarized as *Recipes* of ChEF. We further introduce recipes to assess MLLMs' six dimensions of desiderata and conduct large-scale experiments to test the generalizability of MLLMs across various scenarios and their composite capability for multimodal interactions.

**Limitations.** As one of the pioneering works in this domain, our study has certain limitations. Firstly, ChEF is still in its nascent stage, currently supporting only a limited number of *Scenarios* and models. For instance, *Scenarios* evaluating safety and biases have not been incorporated yet. As we move forward, we aim to include a wider array of *Scenarios* and other models to further enrich and expand the framework's applicability and comprehensiveness. Secondly, there remains a discernible performance variance among models when confronted with different queries. While our provided *Recipes* have significantly mitigated these disparities, such variations are inevitable. Further research is needed to more accurately assess and optimize model performances across diverse queries to achieve more consistent evaluation outcomes. Furthermore, the utilization of the GPT API for evaluation remains an area where the effectiveness has not been conclusively determined. We will continue to stay updated with the latest advancements in the field and leverage the scalability of ChEF to optimize and update accordingly.

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
