# ChEF: A Comprehensive Evaluation Framework for Standardized Assessment of Multimodal Large Language Models (Supplementary Materials)

## Contents

# A  RELATED WORKS

## A.1  MULTIMODAL LARGE LANGUAGE MODELS

Due to the success of large Language models (LLMs) like GPTs (Radford et al., 2019; Brown et al., 2020; Ouyang et al., 2022), LLAMA (Touvron et al., 2023) and Vicuna (Chiang et al., 2023), Multimodal Large Language Models (MLLMs) have recently experienced substantial development. InstructBLIP (Dai et al., 2023), LLaVA (Liu et al., 2023a), and MiniGPT-4 (Zhu et al., 2023) are based on open-source LLMs using vision-language instruction tuning get promising results. mPLUG-Owl (Ye et al., 2023) leverages the capabilities of pre-trained LLMs, a visual knowledge module, and a connected visual abstractor module to effectively align images with text. LAMM (Yin et al., 2023) extend the research of MLLMs to point clouds and propose a training framework optimized for modalities' extension. Otter (Li et al., 2023a) utilizes multimodal context instruction tuning data, demonstrating an improved ability to follow instructions and in in-context learning. LLaMA-Adapter-v2 (Gao et al., 2023) propose an early fusion strategy to solve the interference between image-text alignment and instruction following learning targets. Shikra (Chen et al., 2023a) and Kosmos-2 (Peng et al., 2023) integrate grounding data during the training phase, enabling the model to develop grounding capabilities. In order to comprehensively assess the capabilities of these MLLMs, we present the first Comprehensive Evaluation Framework (ChEF) that can holistically profile each MLLM and fairly compare different MLLMs.

## A.2  BENCHMARKS FOR LARGE LANGUAGE MODELS

In recent years, significant efforts have been made to comprehensively evaluate large language models from diverse perspectives (Liang et al., 2022; Wang et al., 2023d; Bommasani et al., 2021; Gehrmann et al., 2021; 2022; Brown et al., 2020; Gao et al., 2021; von Werra et al., 2022; Srivastava et al., 2022). Gao et al. (2021) provides a unified framework to test autoregressive language models on a large number of different evaluation tasks. Liang et al. (2022) measures seven metrics that reflect a range of societal considerations, including accuracy, calibration, robustness, fairness, bias, toxicity, and efficiency, in order to improve the transparency of language models. Li et al. (2023c) propose to evaluate the instruction following ability from the aspect of how well models can follow instructions that may not align with their priors. Recent studies evaluating the quality of natural language generation (Zheng et al., 2023; Liu et al., 2023b; Wang et al., 2023a) have indicated that GPT-based metrics typically exhibit superior performance compared to traditional reference-based and reference-free baseline metrics in terms of their correlation with human quality judgments. These evaluation metrics effectively assess the capabilities of LLMs from multiple dimensions. However, in the evaluation of MLLMs, there is currently a lack of frameworks and relevant metrics. These frameworks and metrics are of significant importance in assessing MLLMs.

## A.3  BENCHMARKS FOR MULTIMODAL LARGE LANGUAGE MODELS

MLLMs have demonstrated remarkable capabilities (Liu et al., 2023a; Zhu et al., 2023; Dai et al., 2023) and are poised to address increasingly complex multimodal tasks. Various benchmarks have emerged to evaluate MLLMs. Some works focus on evaluating MLLMs using existing conventional multimodal datasets (Wang et al., 2023c) or only evaluate one or a few factors of MLLMs (Shao et al., 2023; Li et al., 2023d; Yu et al., 2023; Bitton et al., 2023), which may not provide a comprehensive evaluation suitable for these models. Recent benchmarks (Li et al., 2023b; Liu et al., 2023c; Fu et al., 2023) often focus on building a multimodal evaluation dataset for MLLMs. These benchmarks have been designed to transform open-ended predictions into predefined categorical choices. For instance, MME transforms free-form responses into binary True/False questions, while Li et al.

(2023b); Liu et al. (2023c) employ multi-choice questions. However, the efficacy of these benchmarks is contingent upon the quality of the dataset construction and may suffer from scalability issues. More recently, efforts such as Yin et al. (2023); Xu et al. (2023) have attempted to establish evaluation frameworks, yet they have been characterized by limitations in terms of scalability and comprehensiveness. In response to these challenges, ChEF offers a standardized framework for conducting versatile evaluations and facilitates seamless integration of new models and tasks.

## B ChEF (Comprehensive Evaluation Framework) Modules

ChEF is a comprehensive evaluation framework aiming at providing a fair and holistic assessment of MLLMs' performance across diverse multimodal tasks. To accomplish this objective, our design principles encompass the following key aspects: Modular, Scalable, Flexible, Reliable, and Indicative. Based on these principles, we carefully design and implement ChEF with four components *i.e.*, *Scenario*, *Instruction*, *Inferencer*, and *Metric*. In this section, we will introduce the details of each module.

### B.1 Scenario

The *Scenario* pertains to the datasets and tasks utilized for evaluating the proficiency of MLLMs in visual and multimodal tasks. Following the principles, the *Scenario* is designed to be scalable. Any *Scenario* can be easily integrated into ChEF by defining the required *Instruction* and *Metric* with the provided interfaces. Due to the substantial similarities among datasets within the same visual task, we categorize them based on task divisions. Within each task, the *Scenarios* can share similar implementations for the given interfaces.

To facilitate the active participation of the open-source community in expanding the scope of *Scenarios*, we incorporate several prominent datasets from highly regarded visual tasks as exemplary *Scenarios*. These datasets include CIFAR-10 (Krizhevsky & Hinton, 2009) for classification, Flickr30k (Young et al., 2014) for image captioning, ScienceQA (Lu et al., 2022) for multimodal question-answering, *etc.* Furthermore, we seamlessly integrate multi-task datasets, including MM-bench (Liu et al., 2023c), SeedBench (Li et al., 2023b), and MME (Fu et al., 2023), into the framework of ChEF. We warmly welcome the integration of additional *Scenarios* into ChEF by simply implementing the requirements with the provided interfaces.

### B.2 Instruction

The *Instruction* component plays a pivotal role in facilitating the model's comprehension of the underlying semantics within the *Scenario* and generating pertinent responses. Within ChEF, a standard query is initially incorporated for each *Scenario*, such as "The photo of" for classification, providing the model with a basis for answer generation. Nevertheless, it is noteworthy that divergent models may interpret the same query dissimilarly, leading to variations in evaluation.

To ensure the universal compatibility of the *Instruction* module, in line with the design principle of flexibility, we undertake measures to devise the query pool, encompassing frequently employed queries that exhibit similar intents. This designation allows for the seamless integration of new queries, thereby ensuring the requisite adaptability for different MLLMs. The standard query and query pool are collectively referred to as `Query`.

Moreover, we firmly believe that leveraging the In-context Example (`ICE`) as the *Instruction* presents a more comprehensive and generalized approach, empowering models to grasp the intricacies of the assigned task and generate responses in the desired format and content. The `ICE` is retrieved from the dataset based on various criteria commonly employed in the field of NLP, including Random `ICE`, Fixed `ICE`, and Top-$K$ `ICE` (Wu et al., 2023; Liu et al., 2022; Su et al., 2023).

**(1) Random ICE** is retrieved at random, without considering their relevance or importance. An example is shown in Figure 1.

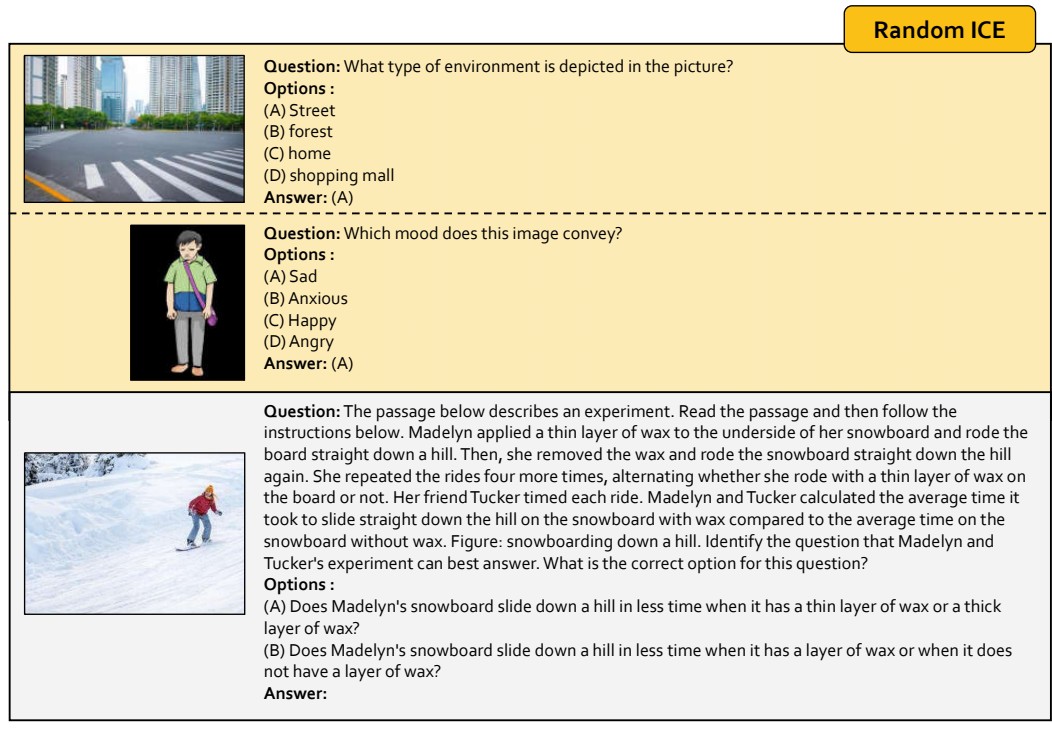

Figure 1: **An example of Random ICE.** The Random `ICE` are randomly retrieved from the dataset, without considering their relevance or importance.

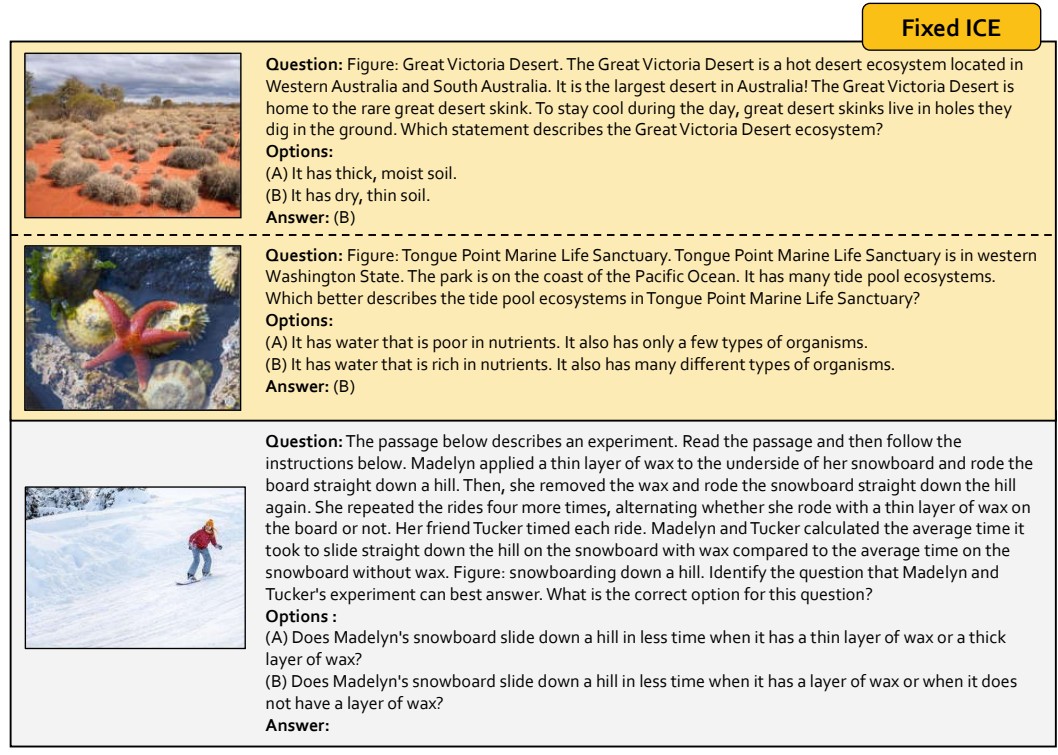

Figure 2: **An example of Fixed ICE.** The Fixed `ICE` is predetermined based on prior knowledge or experiment.

**Top-k Text ICE**

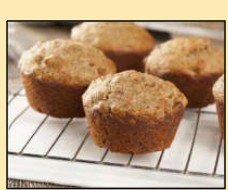

**Question:** The passage below describes an experiment. Read the passage and then follow the instructions below.
Carson made six batches of muffins over the course of one day. He used whole wheat flour in three of the batches and white flour in the other three batches. He divided the batter into muffin tins, using two ounces of batter per muffin. He baked the muffins in a 350¬∞F oven for 20 minutes. After allowing the muffins to cool, Carson measured the dimensions of the muffins and calculated their volumes. He compared the volumes of the muffins made with whole wheat flour to the volumes of the muffins made with white flour. Figure: muffins cooling. Identify the question that Carson's experiment can best answer.
**Options:**
(A) Does the type of flour used in the muffins affect the number of muffins that turn brown after 30 minutes in the oven?
(B) Do muffins made with white flour have larger volumes than muffins made with whole wheat flour?
**Answer:** (B)

- - - - - - - - - - - - - - - - - - - - - - - - - - - - - - - - - - - - - - - - - - - - - - -

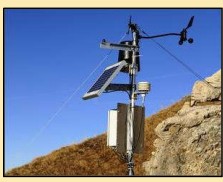

**Question:** People can use the engineering-design process to develop solutions to problems. One step in the process is testing if a potential solution meets the requirements of the design. The passage below describes how the engineering-design process was used to test a solution to a problem. Read the passage. Then answer the question below. Devin was a mechanical engineer who was designing to record temperature, precipitation, and wind speed. The weather station would be used in a town where the highest recorded temperature was 40℃. Devin wanted to make sure the weather station would work even in unusually warm weather. So, he set an indoor test chamber to 50℃ with low moisture and no wind. He left the weather station in the chamber overnight. The next day, he checked to see if the weather station displayed accurate measurements after 24 hours at 50℃. Figure: a weather station. Which of the following could Devin's test show?
**Options:**
(A) if the weather station would work when the temperature was 50℃
(B) how well the weather station would work when it was windy
**Answer:** (A)

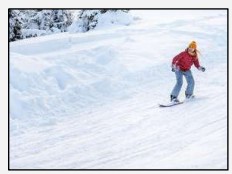

**Question:** The passage below describes an experiment. Read the passage and then follow the instructions below. Madelyn applied a thin layer of wax to the underside of her snowboard and rode the board straight down a hill. Then, she removed the wax and rode the snowboard straight down the hill again. She repeated the rides four more times, alternating whether she rode with a thin layer of wax on the board or not. Her friend Tucker timed each ride. Madelyn and Tucker calculated the average time it took to slide straight down the hill on the snowboard with wax compared to the average time on the snowboard without wax. Figure: snowboarding down a hill. Identify the question that Madelyn and Tucker's experiment can best answer. What is the correct option for this question?
**Options :**
(A) Does Madelyn's snowboard slide down a hill in less time when it has a thin layer of wax or a thick layer of wax?
(B) Does Madelyn's snowboard slide down a hill in less time when it has a layer of wax or when it does not have a layer of wax?
**Answer:**

Figure 3: **An example of Top-$k$ Text ICE.** The Top-$k$ Text ICE is retrieved from the dataset based on text similarity.

**(2) Fixed ICE** is predetermined based on prior knowledge or experiments. These ICE can serve as instructional cues to encourage the model to replicate and generate outputs in a format consistent with the provided examples, as shown in Figure 2

**(3) Top-$k$ ICE** is retrieved based on either the image similarity (Top-$k$ Image ICE) or the text (Top-$k$ Text ICE) similarity, as shown in Figure 3,4.

The designation and implementation of the Query and ICE significantly contribute to the flexibility of evaluation.

### B.3 INFERENCER

The *Inferencer* plays a vital role in determining the model's response to questions. Within ChEF, it incorporates a fundamental auto-regressive generation method. However, due to the free-form and long-term nature of its output, evaluating the quality of the generated text becomes subjective and unreliable (Yin et al., 2023; Li et al., 2023b). To address this concern, we design the following *Inferencers* to support reliable evaluation:

Figure 4: **An example of Top-$k$ Image ICE.** The Top-$k$ Image `ICE` is retrieved from the dataset based on image similarity.

**(1) Direct:** This is an auto-regressive generation method employed without sampling. The output of the MLLMs is determined through greedy search, ensuring consistent output across multiple inference instances for enhanced reliability.

**(2) Chain-of-Thought (CoT):** This answering approach includes a special query, "Let's think step by step", which prompts the model to provide responses in a sequential manner. It prompts the model to provide its reasoning process, ensuring that the model's answers are well-thought-out and dependable.

**(3) Perplexity (PPL):** This *Inferencer* constrains MLLMs' output within a limited text scope, named as answer pool, and derives the answer by computing the likelihood. The answer pool is either fixed, retrieved, or generated based on the specific *Scenario*. For example, in multi-choice question-answering *Scenarios*, the answer pool is the four options {A, B, C, D}. For certain *Scenarios*, it includes the ground-truth answer and several negative candidates either generated or retrieved. `PPL` confines the model's output within a specific range, guaranteeing that the model selects exactly matched answers based on discrimination rather than generating similar responses. Treating MLLMs as discriminative entities for specific *Scenario* evaluation enhances objectivity and reliability in the evaluation process.

**(4) Multi-Turn:** This method decomposes complex tasks into subtasks and generates answers sequentially based on each subtask. For example, in the context of object detection, the initial *Instruction* may pertain to the object categories present in the image, followed by subsequent inquiries regarding the bounding boxes for each detected object category. This approach supports objective and reliable evaluation by assessing the model's responses to each subtask, thereby enhancing objectivity and reliability. Notably, various *Inferencers* can be invoked and seamlessly integrated with one another within multiple turns. For illustration, the `CoT` can be employed during the initial turn, while the subsequent turn can leverage the `Direct`.

These *Inferencers* augment the evaluation framework of ChEF, enabling more objective and trust-worthy assessments of model performance.

## B.4 METRIC

The selection of *Metric*s is crucial when evaluating MLLMs, as it should encompass the evaluation capabilities for traditional visual tasks while considering the novel characteristics of MLLMs as generative models. In the context of traditional computer vision tasks, we believe it is more suitable to conduct adaptation based on the existing evaluation metrics. As a result, within the ChEF framework, we integrate well-established metrics such as BLEU for captioning, accuracy for classification, and mAP for detection, which are commonly used in traditional computer vision tasks.

Additionally, when employing the `PPL` as *Inferencer* in evaluation pipelines, we rely on accuracy as the primary *Metric* since the generated text is confined to an answer pool. This methodology enables the harmonization of evaluation across various *Scenarios*, as accuracy is adopted as the shared assessment criterion.

## C DESIDERATA

Based on ChEF, it becomes rather convenient to set up new evaluations to quantify the desired capabilities (or called **desiderata**) that a competent MLLM model should possess, as a reliable agent that can perform real-world multimodal interactions. The desiderata include calibration, in-context learning, instruction following, language performance, hallucination, and robustness. In this section, we will introduce the details of each desideratum.

## C.1 CALIBRATION

Calibration aims to evaluate the model's performance to be simultaneously accurate and to provide appropriate uncertainty in its outputs, as emphasized in the work by HELM (Liang et al., 2022). This is particularly significant in risk scenarios We evaluate calibration by Expected Calibration Error (ECE) (Naeini et al., 2015; Guo et al., 2017). Formally, let $y$ be the ground truth, and $\hat{y}$ be the model's prediction with associated confidence $\hat{p}$. The ECE examines the difference between the model's predicted confidence $\hat{p}$ and the probability the model is correctly given $\hat{p}$, as shown in equation 1.

$$\text{ECE} = \mathbb{E}[|\hat{p} - \mathbb{E}(y = \hat{y}|\hat{p})|] \tag{1}$$

To estimate the expected accuracy $\mathbb{E}(y = \hat{y}|\hat{p})$ from finite samples, we compute the ECE by binning the model's predictions into $m$ bins following prior work (Guo et al., 2017; Liang et al., 2022). We choose uniform-mass bins for better approximation with $k = 10$, where an equal number of samples fall into each bin. Let $\mathcal{B}_m$ be a set of indices $i$ of samples falling in $m$-th bin, then the average confidence and accuracy of $\mathcal{B}_m$ are defined as

$$\text{conf}(\mathcal{B}_m) = \frac{1}{|\mathcal{B}_m|} \sum_{i \in \mathcal{B}_m} \hat{p}_i \tag{2}$$

$$\text{acc}(\mathcal{B}_m) = \frac{1}{|\mathcal{B}_m|} \sum_{i \in \mathcal{B}_m} \mathbf{1}(\hat{y}_i = y_i) \tag{3}$$

Therefore, we can approximates equation 1 by equation 4.

$$\text{ECE} = \sum_{m=1}^{k} \frac{|\mathcal{B}_m|}{n} |\text{conf}(\mathcal{B}_m) - \text{acc}(\mathcal{B}_m)| \tag{4}$$

The difference between conf and acc for a given bin represents the calibration gap (visualized in Figure 7 ). The lower the ECE, the better the calibration of the model, indicating that the predicted confidence $\hat{p}$ more accurately represents the true probability.

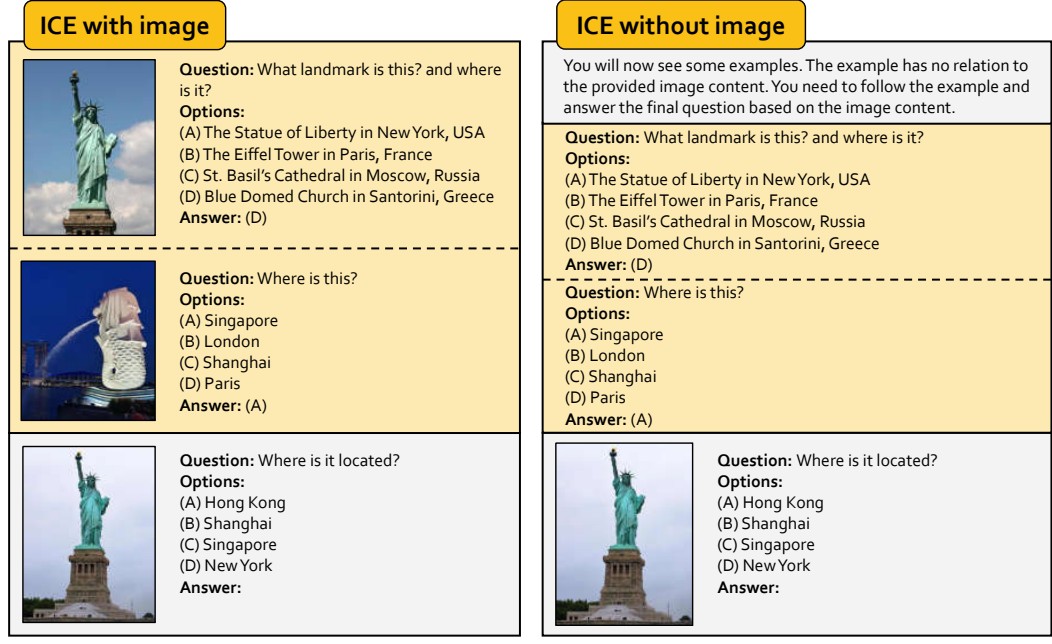

Figure 5: **Difference between ICE with image and without image.** The `ICE` are retrieved based on the images' similarity to the input images.

## C.2   IN-CONTEXT LEARNING

In-context Learning (ICL) aims to evaluate MLLMs' ability to perform new tasks without any gradient-based training (Wu et al., 2023; Brown et al., 2020). This ability is capable of generalizing to unseen cases, which opens up many new technological possibilities that were previously considered unique to humans. While in the field of NLP, LLMs have demonstrated their ability for ICL. However, within the domain of MLLMs, this potential remains unexplored. Most MLLMs lack the ability for ICL (Li et al., 2023a). Therefore, considering the ICL ability is crucial when evaluating multimodal large language models.

ICL adds a small number of `ICE` before `Query` as the *Instruction* and has demonstrated its ability to enhance the performance of LLMs in few-shot scenarios. Given that multimodal tasks typically involve visual data, incorporating the `ICE` with images in MLLMs is a reasonable approach. However, some MLLMs currently only support single-image input. Given the presence of an image in the `Query`, the image of `ICE` cannot be included. Considering the limited support for multi-image input in certain MLLMs, we implement two ICL methodologies: one utilizing `ICE` without image and the other incorporating `ICE` with images, as shown in Figure 5. In the case of `ICE` without image, to prevent any confusion between the content of `ICE` and the images in the `Query` for the MLLMs, we add an additional *Instruction*, explicitly informing the MLLMs that the provided `ICE` text has no relation to the provided image content. For the selection of `ICE`, we implement retriever methods such as Random, Fixed, and Top-$k$, as mentioned in Section B.2.

To measure MLLMs' ICL ability, we utilize `ICE` as *Instruction* for each specific *Scenario*. We compute their accuracy and use the relative accuracy change as the final score. Specifically, we compute the accuracy under the 0-shot setting (without using `ICE`) and the average accuracy values for varying numbers of `ICE`, ranging from 1 to $N$. In multi-choice question-answering paradigms, random guessing can yield an expected lower-bound accuracy, which can be misleading in terms of performance evaluation. To mitigate the impact of this potentially deceptive performance on robustness assessments, we systematically eliminate the bias introduced by random choice. Therefore, we introduce the Relative ICL Accuracy for Multi-choice (RIAM), adapted from Chen et al. (2023b); Schiappa et al. (2022), to more accurately assess the model's ICL ability. The RIAM primarily calculates the relative accuracy change of the model before and after using `ICE`.

## C.3  INSTRUCTION FOLLOWING

Taking inspiration from (Li et al., 2023c), we utilize three groups of instructions for verbalizer manipulation: *natural, neutral, unnatural*, to evaluate how well models can follow instructions that may not align with their priors. The levels in terms of aligning with prior knowledge of these three groups are ranked as *natural > neutral > unnatural*. We expect the model to answer the question following instructions and generate a new answer corresponding to the original answer. In practice, we select different numbers of verbalizers for each group of verbalizer manipulation, depending on the alignment with the model's prior knowledge. Each verbalizer maps "A|B|C|D" to different new options.

**(1) Natural.** "1|2|3|4|5" ,"I|II|III|IV|V" and "first|second|third|fourth|fifth".

**(2) Neutral.** "Smith|Johnson|Williams|Jones|Brown" and "foo|dog|hip|oh|cat".

**(3) Unnatural.** The choices are mapped to their respective next choices as the new verbalizer for each given question (e.g., "D|A|B|C" corresponding to "A|B|C|D").

We calculate the Match Ratio (MR) to determine the percentage of samples that adhere to the verbalizer manipulation instructions, mapping their original answers to corresponding new answers. This calculation helps mitigate the influence of the model's accuracy in answering questions and highlights its proficiency in following verbalizer manipulation instructions. A higher MR indicates a superior ability of the model to follow instructions.

## C.4  LANGUAGE PERFORMANCE

**GPT-4 System Message**

Please act as an impartial judge and conduct a comprehensive assessment of a multimodal AI assistant's performance in the field of Visual Question Answering (VQA). Each data sample to be evaluated follows the following format:

[Question]
{Question}
[Ground Truth]
{Ground truth}
[Assistant's Chain of Thought]
{Chain of thought generated by AI assistant}
[Assistant's Final Choice]
{Final Choice generated by AI assistant}

Your task is to evaluate the quality of natural language generation from AI assistant considering factors such as the helpfulness, relevance, accuracy, depth, creativity, and level of detail of the response.
Please first provide a comprehensive explanation of your evaluation and an overall score ranging from 0 to 10 based on explanation, where a higher score indicates better overall performance. Please output in the following format:

[Explanation]
{Evaluation Explanation}
[Overall Score]
{An integer ranging from 0 to 10 representing the final evaluation score}

Please ensure that your evaluation score comprehensively captures the AI assistant's performance avoiding any potential bias. Assuming that the visual information mentioned by the AI assistant is contained in the image, you only need to evaluate the quality of the generated text. Your assessments will contribute to enhancing the assistant's effectiveness in visual question answering.

Figure 6: **System message for GPT-4** to evaluate language performance of MLLMs. The System Message includes the evaluation task description, the format of the evaluation input template, the evaluation criteria, and the format of the evaluation output template. The phrases enclosed in "[]" represent domain names, which remain constant during the testing process. The phrases enclosed in "{}" represent the meanings of the domain names, which is a placeholder to be replaced with the specific content corresponding to the domain name during testing.

Evaluating the quality of natural language generation is a challenging task, often requiring scoring based on various aspects such as coherence, consistency, fluency, and more. Recent studies (Zheng et al., 2023; Liu et al., 2023b; Wang et al., 2023a) have indicated that GPT-based metrics typically exhibit superior performance compared to traditional reference-based and reference-free baseline metrics in terms of their correlation with human quality judgments. Thus, we employ GPT to score

the chain-of-thought text generated by the model in the multimodal question-answering *Scenarios*, aiming to evaluate the model's language performance.

In contrast to NLP, where GPT can evaluate the quality of natural language generation without references (Zheng et al., 2023; Liu et al., 2023b; Wang et al., 2023a), the evaluation process in the visual *Scenarios* presents a distinct challenge as GPT lacks access to visual information. Therefore, we implement specific adaptations for evaluating GPT's performance in multimodal tasks as follows:

**(1) Reference-Based Evaluation:** We provide GPT with ground-truth sentences (*i.e.* answers and questions) as the reference during the evaluation, which ensures faithfulness of the chain-of-thought.

**(2) Visual Information Assumption:** GPT is prompted to assume that all visual information mentioned in the test model's responses is contained in the image. This measure prevents GPT from misjudging descriptions of images in the chain-of-thought as language hallucinations (which may not be explicitly stated in the given question). This helps avoid unwarranted reductions in the language performance score.

**(3) Selective Sampling of Correct Conclusions:** We selectively extract samples in which the MLLMs' conclusions are correct. This reduces the impact of conclusion accuracy on the evaluation of language generation quality, as mentioned in Section E.4.

**(4) Efficient and Scalable Evaluation**: For more efficient and scalable evaluation, instead of pairwise comparisons, we individually assess each MLLM's response, which is called Single Answer Grading. This method exhibits high agreement with human experts in NLP tasks as demonstrated in (Zheng et al., 2023).

**(5) Multiple Evidence Calibration:** (Wang et al., 2023b) To make the GPT score more reliable and interpretable, we prompt the GPT to generate an explanation as evaluation evidence before generating the final overall score. Thanks to the properties of autoregressive models, this method allows GPT to calibrate scores based on evaluation evidence. To further reduce the systematic error of GPT evaluation, we conduct Multiple Evidence Calibration, sampling multiple GPT responses for each evaluation query, and taking the average score of all responses as the final evaluation score.

To apply the adaptations below, we modify the system message for GPT-4. Figure 6 shows the system message for GPT-4 to evaluate the language performance of MLLMs.

## C.5 ROBUSTNESS

Robustness aims at evaluating the capability of MLLMs to maintain accurate performance and meaningful outputs in the face of diverse challenges and variations in input data. This includes addressing data corruption and perturbations, which ensures the model's reliability in real-world applications. To evaluate the robustness of our model, we carefully select mild image and text corruptions, drawing inspiration from recent work (Liang et al., 2022; Qiu et al., 2022; Chen et al., 2023b; Schiappa et al., 2022).

For image corruptions, we incorporate five corruption categories: *noise, blur, weather, digital* (sourced from ImageNet-C (Hendrycks & Dietterich, 2019)), and *others* (fundamental data augmentation techniques). For text corruption, we introduce five categories like (Chen et al., 2023b): *basic, sentence, word, character* (sourced from (Wang et al., 2021)) and *choice*. The *choice* category specifically represents additional corruption introduced for multi-choice question-answering *Scenarios*. All the corruption methods are shown in Table 1 and Table 2. These corruption methods we employ do not distort the core information of the images and text. For instance, the Center Crop for images retains at least 90% of the image content. Text perturbations solely target the questions, and in the options section, only Circular Option and Reverse Option (circular shifting and reverse order on options respectively) are applied, ensuring that the original meaning of the questions and correct answers remain unchanged.

To simulate real-world complexity, we construct composite corruption sequences with random severity levels for both image and text within each sample. Specifically, corruption methods from various categories are composited in a specific order. For each category, the corruption method to apply is selected based on a composite strategy. We employ two strategies: *Random*, where one corruption method from the category is chosen randomly, and *Sequential*, where all methods from the category are applied sequentially. This approach enables us to assess the model's robustness in a scalable

Table 1: **Image corruption methods** are categorized into five types. In the robustness experiments, the corruption for each image is formed by sequentially combining methods each with random severity level from the following five categories: *Noise*, *Blur*, *Weather*, *Digital*, and *Other*. Each category's method is selected based on the corresponding combination strategy: *Random* denotes the random selection of one method from all methods within that category, while *Sequential* implies the consecutive execution of all methods within that category. Severity represents the number of adjustable severity levels for the corruption method.

| Category | Method | Severity | Compose Strategy |
|---|---|---|---|
| **Noise** | Gaussian Noise | 5 | Random |
| | Shot Noise | 5 | |
| | Impulse Noise | 5 | |
| | Speckle Noise | 5 | |
| **Blur** | Defocus Blur | 5 | Random |
| | Frosted Glass Blur | 5 | |
| | Motion Blur | 5 | |
| | Zoom Blur | 5 | |
| | Gaussian Blur | 5 | |
| **Weather** | Snow | 5 | Random |
| | Frost | 5 | |
| | Fog | 5 | |
| | Brightness | 5 | |
| | Spatter | 5 | |
| **Digital** | Contrast | 5 | Random |
| | Elastic | 5 | |
| | Pixelate | 5 | |
| | JPEG Compression | 5 | |
| | Saturate | 5 | |
| **Other** | Center Crop | 5 | Sequential |
| | Resize | 5 | |
| | Rotate | 5 | |

Table 2: **Text corruption methods** are categorized into five types. In the robustness experiments, the corruption for each text is formed by sequentially combining methods each with random severity level from the following five categories: *Basic*, *Sentence*, *Word*, *Character*, and *Choice*. Each category's method is selected based on the corresponding combination strategy: *Random* denotes the random selection of one method from all methods within that category, while *Sequential* implies the consecutive execution of all methods within that category. Severity represents the number of adjustable severity levels for the corruption method.

| Category | Method | Severity | Compose Strategy |
|---|---|---|---|
| **Basic** | Lowercase | 1 | Sequential |
| | Constraction/Expansion | 1 | |
| **Sentence** | Passive | 1 | Random |
| | Active | 1 | |
| | Casual | 1 | |
| | Formal | 1 | |
| | Back Translation | 1 | |
| **Word** | Swap Synonym | 5 | Random |
| | Insert Adv. | 1 | |
| | Add Irrelevant | 1 | |
| **Character** | Ocr | 5 | Random |
| | Typos | 5 | |
| | Spelling Error | 5 | |
| | Keyboard | 5 | |
| **Choice** | Circular Options | 1 | Random |
| | Reverse Options | 1 | |

manner, rather than evaluating the model for each instance of every separate corruption. By applying image corruption and text corruption at the same time, we can evaluate the model's performance in handling joint corruption across visual and textual domains.

To assess the model's robustness more accurately, we introduce the Relative Robustness for Multi-choice (RRM). Similar to the RIAM described in Section C.2, we eliminate the bias introduced by random choice. The RRM primarily calculates the relative accuracy change of the model beyond random guessing accuracy before and after corruptions.

### C.6 HALLUCINATION

Hallucination refers to the generated content that is nonsensical or unfaithful to the provided source content (Ji et al., 2023). Similar to LLMs, MLLMs also encounter the challenge of hallucination. Since objects are the core elements that contribute to the visual semantics of an image, we study the object hallucination problem, which refers to the generated descriptions containing objects that are inconsistent with the given image (Biten et al., 2022). As a result, we utilize the Polling-based Object Probing Evaluation (POPE) pipeline (Li et al., 2023d) on MSCOCO (Lin et al., 2014). The fundamental concept behind this approach is to transform the evaluation of hallucination into a series of binary classification tasks. This is achieved by presenting MLLMs with straightforward Yes-or-No questions regarding the presence of specific objects within the images (e.g., "Is there a car in the image?"). Each image is prompted with six such Yes-or-No questions. To generate the probing objects, POPE considers three polling strategies by sampling the objects randomly, from popular objects, and among those frequently co-occurring objects, respectively. Additionally, we employ `PPL` to enhance the reliability of our evaluation. Similar to POPE, we also adopt *Metrics* including accuracy, precision, recall, F1-Score, and the ratio of "Yes" responses.

## D  EXPERIMENTS: DETAILS OF EVALUATION SETUP

### D.1  DETAILS OF THE EVALUATED MODELS

Table 3: **Details of the evaluated MLLMs.** mPLUG stands for mPLUG-Owl and LAv2 stands for LLaMA-Adapter-v2.

| MLLM | Visual Model | Language Model | Overall Parameter |
|---|---|---|---|
| **LLaVA** | CLIP ViT-L/14 | MPT 7B | 7B |
| **LAMM** | CLIP ViT-L/14 | Vicuna 13B | 13B |
| **MiniGPT-4** | EVA-G | Vicuna 7B | 8B |
| **mPLUG** | CLIP ViT-L/14 | LLaMA 7B | 7B |
| **Otter** | CLIP ViT-L/14 | LLaMA 7B | 9B |
| **LAv2** | CLIP ViT-L/14 | LLaMA 7B | 7B |
| **InstructBLIP** | EVA-G | Vicuna 7B | 8B |
| **Shikra** | CLIP ViT-L/14 | LLaMA 7B | 7B |
| **Kosmos-2** | CLIP ViT-L/14 | Decoder 1.3B | 1.6B |

Table 4: **Success rate in choice extraction on MMBench.** The results represent the success rate in choice extraction of Step-1, which is defined in MMBench. MMBench released the evaluation code for three models. The results in ChEF are aligned with those in MMBench.

|  | MMBench | ChEF |
|---|---|---|
| LLaVA | 14.85 | 14.78 |
| MiniGPT-4 | 55.58 | 52.52 |
| InstructBLIP | 91.2 | 91.52 |

In Table 3, we show the details of all the evaluated MLLMs in ChEF. In order to ensure that the evaluated MLLMs are relatively up-to-date, we attempt to align the results of the choice extraction success rate in Step-1 with MMBench (Liu et al., 2023c), which is a recently proposed multimudal benchmark. We align the results with all the open-sourced evaluated MLLMs in MMBench, as

Table 5: **Details of default *Recipes*.** Acc. is accuracy. `CoT → PPL` means `Multi-Turn` with `CoT` in the first turn and `PPL` in the second.

| Scenario | Instruction | Inferencer | Metric |
|---|---|---|---|
| **CIFAR10** | Standard Query | PPL | Acc. |
| **Omnibenchmark** | Standard Query | Multi-Turn PPL | WeightedACC |
| **Flickr30k** | Standard Query | PPL | Acc. |
| **VOC2012** | Standard Query | Multi-Turn PPL | Acc. |
| **FSC147** | Standard Query | PPL | Acc. |
| **ScienceQA** | Standard Query | CoT → PPL | Acc. |
| **MMBench** | Standard Query | CoT → PPL | Acc. |
| **MME** | Standard Query | PPL | Acc. |
| **SEEDBench** | Standard Query | PPL | Acc. |

Table 6: **Results of VanillaEval and CircularEval on MMBench.** The results reveal a substantial decrease in accuracy when switching from VanillaEval to CircularEval.

| | VanillaEval | CircularEval |
|---|---|---|
| **LLaVA** | 43.13 | 10.24 |
| **LAMM** | 44.47 | 14.21 |
| **MiniGPT-4** | 54.34 | 26.46 |
| **mPLUG** | 49.57 | 12.24 |
| **Otter** | 53.91 | 26.27 |
| **LAv2** | 57.06 | 24.01 |
| **InstructBLIP** | 65.73 | 46.8 |
| **Shikra** | 63.26 | 43.08 |
| **Kosmos-2** | 25.60 | 0.1 |

shown in Table 4. Due to differences in evaluation settings, such as input queries, inference strategies, and metrics, the evaluated results on MMBench in ChEF may differ slightly from those in MMBench.

## D.2 DEFAULT RECIPES FOR SCENARIOS

In ChEF, we provide default *Recipes* for each *Scenario*. In Table 5, we show the details of the default *Recipes* for each *Scenario*. Among the *Scenarios*, the Omnibenchmark is meticulously labeled using a hierarchical chain of categories, facilitated by the Bamboo tree methodology (Zhang et al., 2022). For *Instruction*, we employ standard queries as nearly all MLLMs lack the ability for in-context learning.

For *Inferencer*, we adopt `PPL` for most *Scenarios*. For ScienceQA and MMBench, we employ `Multi-Turn`, with the first turn using the `CoT`, followed by the `PPL` in the second turn. For fine-grained classification tasks, we utilize the `Multi-Turn`, where each turn is a `PPL`, to hierarchically inquire about categories. For detection tasks, the first turn employs `PPL` to inquire about categories, while the second turn utilizes `PPL` to inquire about bounding boxes. The answer pool for CIFAR-10 encompasses the ten predefined classes, while for FSC147, it involves the ground truth values with an additional range of ±2. The answer pool for Omnibenchmark is randomly retrieved from the category tree in Bamboo (Zhang et al., 2022). In the case of Flickr30k, the answer pool is determined by retrieving the top-$k$ negative candidates from the test data based on BERT similarity (Reimers & Gurevych, 2019). The answer pool for VOC2012 is randomly generated by scaling and translating the ground-truth bounding boxes. The answer pool for multimodal question-answering tasks is the options {A, B, C, D}.

In the *Metric*, a single accuracy measure is utilized to assess all *Scenarios* uniformly. For certain specialized *Scenarios*, we adopt specific approaches to calculate accuracy. For Omnibenchmark, weighted accuracy is employed, which entails a weighted accuracy calculation based on the granularity of the predicted classification. MMBench provides two evaluation settings (*i.e.*, VanillaEval and CircularEval), where the CircularEval is used to assess the MLLMs' consistency in responses for the same question when the order of options is changed. We conduct evaluations in both settings,

Table 7: **Details of Recipes for six dimensions of desiderata.** ICL is in-context learning. Ins. Follow. is instruction following and Lang. Perf. is language performance.

| Desiderata | Scenario | Instruction | Inferencer | Metric |
|---|---|---|---|---|
| Calibration | MMBench + ScienceQA | Standard Query | CoT → PPL | ECE |
| ICL | MMBench + ScienceQA | Random ICE | CoT → PPL | RIAM |
| Ins. Follow. | MMBench + ScienceQA | Standard Query | CoT → PPL | MR |
| Lang. Perf. | ScienceQA | Standard Query | CoT → PPL | GPT-based Metric |
| Robustness | MMBench + ScienceQA | Standard Query | CoT → PPL | MRR |
| Hallucination | MSCOCO | Standard Query | PPL | Acc |

Table 8: **Results of calibration.** Acc. stands for accuracy and ECE is the Expected Calibration Error. The overall score is calculated through 1 - weighted average ECE, representing the reliability of the model's prediction probability. The entries that are both bold and underlined indicate the best performance.

| Scenario
MLLM | ScienceQA | | MMBench | | Overall |
|---|---|---|---|---|---|
| | Acc. ↑ | ECE% ↓ | Acc. ↑ | ECE% ↓ | |
| LLaVA | 46.55 | **7.26** | 44.13 | 14.66 | 90.01 |
| LAMM | 52.75 | 20.79 | 44.47 | 28.52 | 76.36 |
| MiniGPT-4 | 47.00 | 15.28 | 54.34 | 15.24 | 84.73 |
| mPLUG | 48.44 | 15.72 | 49.57 | 15.47 | 84.15 |
| Otter | 50.22 | 21.10 | 53.91 | 10.52 | 82.80 |
| LAv2 | 54.34 | 8.17 | 57.06 | 14.19 | 89.61 |
| InstructBLIP | **55.18** | 10.57 | **65.73** | **6.25** | **91.25** |
| Shikra | 45.21 | 14.57 | 63.26 | 6.65 | 88.35 |
| Kosmos-2 | 34.60 | 10.63 | 25.60 | 11.13 | 89.19 |

as shown in Table 6. Across all MLLMs, a significant decline is observed, indicating MLLMs' poor performance in consistency. The utilization of CircularEval assesses a composite capability with both visual performance and consistency. To disentangle these two dimensions of capability, we employ the VanillaEval for the default *Recipe* and incorporate hallucination and robustness within the desiderata to evaluate the dimensions associated with consistency.

### D.3 RECIPES FOR DESIDERATA

We employ specialized *Recipes* to assess the six dimensions of desiderata, as shown in Table 7. All the six dimensions of desiderata except language performance and hallucination are evaluated on MMBench and ScienceQA. Language performance is evaluated on 250 samples random retrieved from ScienceQA and MMBench. Following POPE (Li et al., 2023d), hallucination is specifically assessed on the MSCOCO dataset (Lin et al., 2014).

In terms of the *Instruction*, Random `ICE` is employed as the *Instruction* for ICL evaluation, while standard queries are utilized for the other dimensions. For most MLLMs that lack support for multi-image input, the Random `ICE` consists solely of text, while for MLLMs that do support multi-image input, such as Otter (Li et al., 2023a), the Random `ICE` is adapted to incorporate images. For instruction following evaluation, we concatenate instructions from different groups of verbalizer manipulation at the end of the standard query.

For the *Inferencer*, we employ `Multi-Turn` with the first turn using the `CoT`, followed by `PPL`. The *Metric* we use for each dimension is discussed in Section C.

## E EXPERIMENTS: EMPIRICAL EXPERIMENTS ON DESIDERATA

### E.1 CALIBRATION

The calibration results are presented in Table 8. To illustrate the differences in calibration performance, we also provide reliability diagrams for LLaVA and Otter on ScienceQA in Figure 7. In reliability diagrams, predictions are sorted based on the MLLMs' confidence scores, and an equal number of predictions are grouped into 10 bins. By calculating the average confidence and accuracy

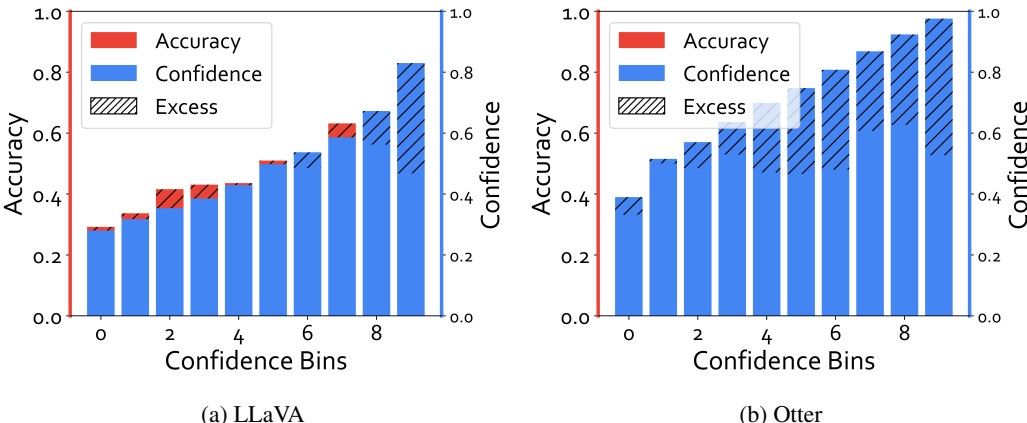

(a) LLaVA                                              (b) Otter

Figure 7: **Reliability diagrams for LLaVA and Otter on ScienceQA.** The red excess parts represent the degree of insufficient confidence of the model, and the blue excess parts represent the degree of overconfidence of the model.

within each bin, we can compare and evaluate the gap between confidence and accuracy intuitively. The observations are as follows:

**(1)** Higher accuracy does not imply better calibration. In ScienceQA, LLaVA demonstrates an average accuracy with the lowest ECE, showing a relatively better calibration. In contrast, Otter achieves higher accuracy with the highest ECE, showing a relatively worse calibration. Reliability diagrams provide a more intuitive and detailed illustration. We can observe that the confidence and actual accuracy in the first 9 bins exhibited a clear correlation, indicating that the predicted confidence of the first 90% of LLaVA is relatively well calibrated. However, the reliability diagram of Otter shows a larger gap between confidence and accuracy, suggesting that Otter's predicted confidence is relatively poorly calibrated.

**(2)** Higher confidence does not imply higher accuracy and better calibration. In the reliability diagrams, both MLLMs have a substantial gap between confidence and accuracy in the last bin, which contains samples with top 10% confidence. Both MLLMs exhibit overconfidence in these samples, which reminds us to avoid considering higher confidence as evidence for higher accuracy. Additionally, it can be observed that the gap between accuracy and confidence does not decrease with increasing confidence, indicating that confidence cannot effectively represent reliability.

**(3)** InstructBLIP achieves the highest accuracy in both visual tasks, while simultaneously exhibiting remarkably low ECE, indicating exceptional calibration. Conversely, other models demonstrate a certain trade-off between the two dimensions. It implies that InstructBLIP can yield superior calibration, so as to provide precise answers to questions while accurately conveying its uncertainty.

### E.2 IN-CONTEXT LEARNING

The evaluations of in-context learning (ICL) are conducted on ScienceQA and MMBench, with `ICE` numbers set at 0, 1, 2, and 3 respectively. The ICL retriever used in the experiments is Random. The experimental results are illustrated in Figure 8(a). To evaluate the influence of accompanying images in `ICE`, we also conduct experiments using Otter, mPLUG-Owl, and MiniGPT-4, as shown in Figure 8(b). These models are evaluated on MMBench using random retrieved `ICE` with and without images respectively. To compare the different performance of MLLMs with retrieved `ICE` under different settings, we further evaluate MMBench, utilizing LLaVA, Shikra, Otter, and MiniGPT-4, as shown in Figure 9. The methodologies employed for the `ICE` retriever include Random, Fixed, Top-$k$ Text, and Top-$k$ Image. The observations are as follows:

**(1)** It can be observed from Figure 8(a) that most of the MLLMs exhibited a decline in performance compared to the zero-shot setting, except for Otter and Kosmos-2. This can be attributed to Otter's training on in-context instruction tuning data, thus enhancing its ICL capabilities. In contrast, the observed improvement in Kosmos-2's performance is due to its struggles to comprehend the mean-

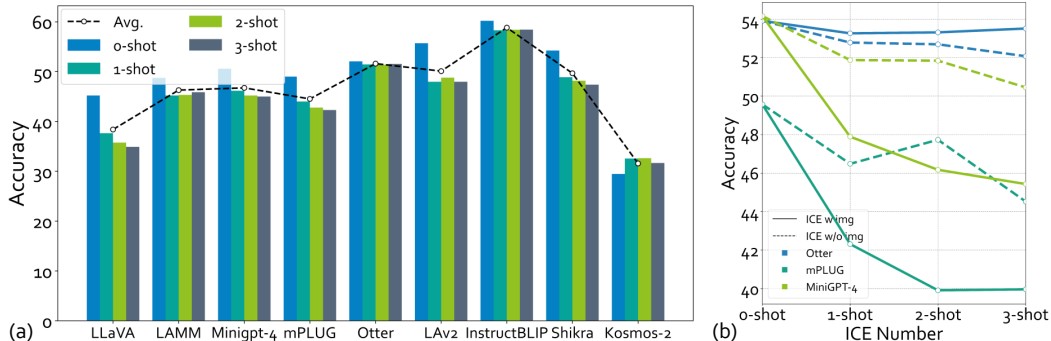

Figure 8: **Results of in-context learning.** (a) Average results of in-context learning on ScienceQA and MMBench utilizing various `ICE` numbers. (b) Results of in-context learning on MMBench for Otter, mPUG-Owl, and MiniGPT-4, utilizing various `ICE` numbers with and without images respectively.

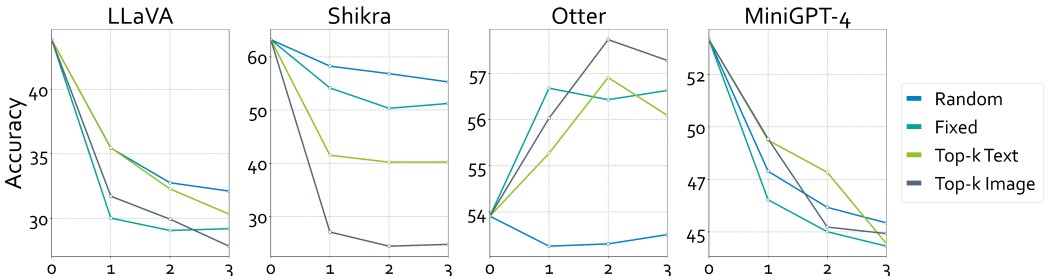

Figure 9: **Experimental results of evaluation with ICE as *Instruction* under different retriever settings.** The retriever methodologies employed encompass Random, Fixed, Top-$k$ Text, and Top-$k$ Image.

ing of options {A, B, C, D} provided in the question, resulting in difficulty in aligning the answers to options. The number of `ICE` does not present a significant impact on the results. From an overall perspective, the majority of MLLMs do not demonstrate capabilities in ICL.

**(2)** Otter demonstrates a slight enhancement when deploying `ICE` with images compared to the `ICE` without image, as shown in Figure 8(b). However, its performance attenuates in the absence of images, failing to manifest its ICL capabilities. This suggests that integrating `ICE` with an image is a judicious design choice within MLLMs. Contrarily, neither mPLUG-Owl nor MiniGPT-4 shows improvement in their capabilities regardless of the presence or absence of images in the `ICE`.

**(3)** It can be observed from Figure 9 that different retrievers have different results, and the Top-$k$ method exhibits slightly inferior performance compared to the others. This potential decline in performance might be attributed to the fact that the MLLMs might regard the given answer in a similar `ICE` as the correct answer for the `Query`, thereby influencing the model's prediction.

### E.3 INSTRUCTION FOLLOWING

Table 9 reports the results of instruction following on ScienceQA and MMBench. We also report the original accuracy Acc and the weighted average accuracy $Acc_{vm}$ of different verbalizer manipulation instructions. To further explore the instruction following, we show the results of different verbalizer manipulations in Figure 10. We also provide the results in Figure 10, that follow the ranking of different groups of instructions in alignment with prior knowledge (*natural > neutral > unnatural*), where MR also decreases sequentially. The observations are as follows:

**(1)** We observed that some MLLMs do not experience a significant decrease in $Acc_{vm}$ compared to Acc when the MR is low. This can be attributed to cases where the original response is incorrect but become correct after verbalizer manipulation. On the other hand, questions that are initially

Table 9: **Results of instruction following.** The abbreviations we use are: Acc for original accuracy; $Acc_{vm}$ for the weighted average accuracy for different instructions of verbalizer manipulation; MR for the weighted average match ratio for different instructions of verbalizer manipulation, as defined in Section C; Avg. for an average of results on ScienceQA and MMBench. The entries that are both bold and underlined indicate the best performance.

| Scenario | ScienceQA | | | MMBench | | | Avg. | | |
| MLLM | Acc ↑ | $Acc_{vm}$ ↑ | MR% ↑ | Acc↑ | $Acc_{vm}$ ↑ | MR% ↑ | Acc ↑ | $Acc_{vm}$ ↑ | MR% ↑ |
|---|---|---|---|---|---|---|---|---|---|
| **LLaVA** | 46.55 | 41.10 | **46.23** | 44.13 | 35.02 | 39.60 | 45.66 | **38.86** | 43.79 |
| **LAMM** | 52.75 | 41.41 | 42.41 | 44.47 | 34.11 | 34.72 | 49.70 | 38.72 | 39.58 |
| **MiniGPT-4** | 47.00 | 36.95 | 43.01 | 54.34 | 41.81 | **43.78** | 49.70 | 38.74 | 43.29 |
| **mPLUG** | 48.44 | 39.93 | 40.28 | 49.57 | 35.39 | 33.43 | 48.86 | 38.25 | 37.76 |
| **Otter** | 50.22 | 38.65 | 38.30 | 53.91 | 33.29 | 36.90 | 51.58 | 36.67 | 37.78 |
| **LAv2** | 54.34 | **41.71** | 44.40 | 57.06 | 27.38 | 28.83 | 55.34 | 36.43 | 38.66 |
| **InstructBLIP** | **55.18** | 38.23 | 45.07 | **65.73** | **37.59** | 43.46 | **59.07** | 38.00 | **44.47** |
| **Shikra** | 45.21 | 35.80 | 37.89 | 63.26 | 31.58 | 32.91 | 51.86 | 34.24 | 36.05 |
| **Kosmos-2** | 34.60 | 35.36 | 17.70 | 25.60 | 32.17 | 14.19 | 31.29 | 34.18 | 16.41 |

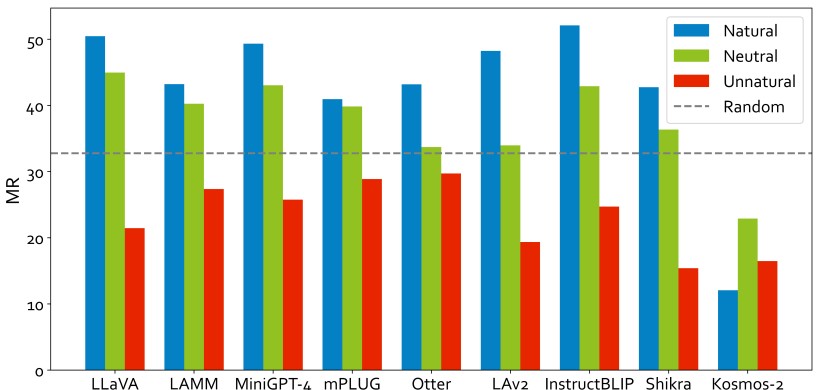

Figure 10: **Results of instruction following with different verbalizer manipulation**, where Natural represents the accuracy with instructions of natural verbalizer; Neutral represents the accuracy with instructions of neutral verbalizer; Unnatural represents the accuracy with instructions of unnatural verbalizer; the dotted line represents the accuracy of random guessing.

answered correctly remain largely consistent between before and after verbalizer manipulation. This suggests that the models exhibit higher instruction following ability on confident questions but are more susceptible to disturbance on uncertain questions. It indicates a correlation between instruction following and confidence in question answering.

**(2)** The distributions of Acc and MR are entirely different, where the distribution of MR is more discriminative. For Kosmos2, $Acc_{vm}$ increases because its original accuracy is lower than that of random guessing (35.80 on ScienceQA, 27.57 on MMBench), and random guessing improves accuracy. Kosmos-2 exhibits a similar $Acc_{vm}$ to Shikra but has the lowest MR, further confirming that Kosmos-2 has degenerated into random guessing, leading to its poor performance on instruction following.

**(3)** The results of natural and neutral are significantly higher than that of unnatural, suggesting that the model is more likely to follow instructions in the natural and neutral categories. This is further supported by their probabilities being higher than random guessing, indicating that the model indeed possesses a certain level of understanding of these two sets of instructions rather than making random guesses. On the other hand, most of the unnatural instructions perform well below the level of random guessing, demonstrating that following unnatural instructions is highly challenging for current MLLMs.

**(4)** InstructBLIP performs best in the natural category but exhibits a noticeable performance drop in the neutral category, suggesting that InstructBLIP relies more on prior knowledge for instruction understanding rather than comprehending new instruction content.

## E.4 LANGUAGE PERFORMANCE

| MLLM | Lang. Perf. |
|------|-------------|
| **LLaVA** | 84.82 |
| **LAMM** | 79.08 |
| **MiniGPT-4** | 70.66 |
| **mPLUG** | 88.44 |
| **Otter** | 74.05 |
| **LAv2** | **90.85** |
| **InstructBLIP** | 80.01 |
| **Shikra** | 66.67 |
| **Kosmos-2** | 45.86 |

(a)

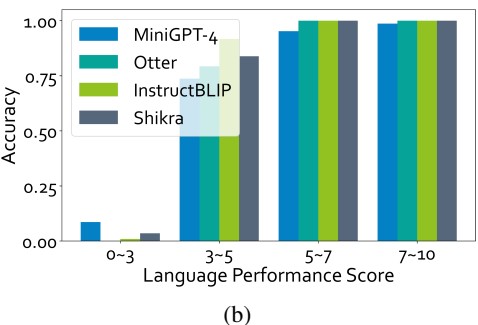

(b)

Figure 11: **Analysis of language performance.** (a) Complete results of language performance. This desiderata only evaluates the natural language generation quality of thought chains in which the model provides correct conclusions to prevent conclusion accuracy from dominating the language performance score. (b) Accuracy distribution in language performance. We divide the evaluation samples into four intervals based on GPT scores and calculate the conclusion accuracy within each interval.

Table 11(a) provides the results of language performance. To illustrate the role of selective sampling of correct, Figure 11(b) displays the distribution of accuracy across different score ranges of language performance scores. We also provide a typical example in Figure 12, comparing the disparity in language performance between LLaVA and Shikra when both provide correct answers. The following presents our key observations:

**(1)** Kosmos-2 exhibits poor performance due to its inability to provide reasoning processes in practical multi-choice question-answering *Scenarios*. Conversely, Shikra demonstrates relatively weak performance attributed to its incapacity to deliver reasoning analysis. Despite prompts intended to elicit the reasoning process, Shikra tends to provide direct answers, leading to lower scores. In real interactive scenarios, MLLMs should offer some form of reasoning alongside their answers. Merely achieving higher accuracy does not necessarily guarantee enhanced interactivity. Therefore, the significance of language performance is highlighted. This finding further emphasizes the imperative need to evaluate language performance in MLLMs.

**(2)** As shown in Figure 11(b), it can be observed that samples with lower language performance scores (0-3) provided by GPT are predominantly incorrect answers by the MLLMs, indicating that the low scores are largely influenced by the accuracy of the answers rather than language performance. Conversely, in the three bins with scores > 3, the accuracy significantly improved. It is worth noting that despite most models providing correct answers, there are substantial differences in language performance. Therefore, it is appropriate to evaluate the language performance of only the correct samples to mitigate the impact of answer accuracy on the evaluation.

**(3)** Figure 12, as a typical example, illustrates the difference in language performance between LLaVA and Shikra, when they both provide correct answers. Regarding Shikra, GPT-4 noted that its generated-chain of-thought, while yielding the correct answer, lacks relevance to the given options. This inconsistency could potentially cause confusion, resulting in a lower score of 6. In the case of LLaVA, its generated chain-of-thought showcases a logical process that adeptly employs visual information for reasoned deductive reasoning. GPT successfully acknowledges the strengths of LLaVA's chain-of-thought, providing a comprehensive explanation for its impressive score of 9. The deduction of one point may be attributed to a limited presence of divergent associations and generalizations.

**Shikra's Language Performance**

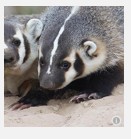

[Question]
Question: Which animal's feet are also adapted for digging?
Options:
(A) groundhog
(B) bottlenose dolphin
[Ground Truth]
(A)

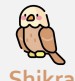
Shikra

[Assistant's Chain of Thought]
two small black and white animals resting on the sand So the answer is badger
[Assistant's Final Choice]
(A)

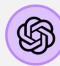
GPT-4

[Explanation]
The assistant did provide the correct answer, which implies some level of accuracy. However, the chain of thought mentioned does not directly connect to the final choice. The assistant indicates two small black and white animals resting on the sand but then proceeds to say the answer is a badger, which is not among the provided options. This inconsistency in the chain of thought may lead to confusion for the user. Regardless, the assistant ultimately provided the correct answer but its line of reasoning wasn't explicitly clear and did not align with the provided options. The assistant could have directly related the image to a groundhog's known ability to dig, providing a more accurate, relevant, and detailed response.

[Overall Score]
6

**LLaVA's Language Performance**

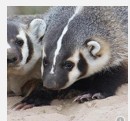

[Question]
Question: Which animal's feet are also adapted for digging?
Options:
(A) groundhog
(B) bottlenose dolphin
[Ground Truth]
(A)

LLaVA

[Assistant's Chain of Thought]
The animal's feet that are adapted for digging are the groundhog's feet. The image shows two groundhogs, one with its feet on the ground and the other with its feet in the air. This indicates that the groundhog's feet are well-suited for digging and burrowing, which is a natural behavior for groundhogs.
[Assistant's Final Choice]
(A)

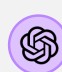
GPT-4

[Explanation]
The AI assistant's answer is helpful, relevant and accurate in terms of the question asked. The assistant correctly identified the groundhog's feet as being adapted for digging, which is the correct answer as provided by the ground truth. Furthermore, the assistant's inference based on the visual clues in the image - "one with its feet on the ground and the other with its feet in the air" - demonstrates an adequate depth of understanding of the scenario and the natural behavior of this particular animal. The method of deduction and articulation the AI utilized was detailed and appropriate for the context of the question, showing however no special creativity. This showcases the strengths of a multimodal AI in its ability to synthesize visual and contextual data for accurate interpretation.

[Overall Score]
9

Figure 12: **Examples of language performance evaluation on Shikra and LLaVA**, where two models exhibit varying levels of natural language generation quality. GPT-4 generates an evaluation explanation as evidence and then generates an overall score based on the Chain-of-Thought and the final choice generated by the Assistant. Here, we present only one explanation and the overall score generated by GPT-4. Note that in practice, for each sample, GPT-4 generates five explanations and their corresponding overall scores through sampling. The final score for the sample is obtained by averaging these five overall scores.

## E.5 ROBUSTNESS

The robustness experiment is conducted on ScienceQA and MMBench, results are presented in Table 10. The $\text{acc}_{\text{random}}$ on ScienceQA is 35.80. The $\text{acc}_{\text{random}}$ on MMBench is 27.57. In order to evaluate the influence of different corruptions, we conduct experiments on ScienceQA and MMBench using different corruption types, as shown in Figure 13. These corruptions encompass both image corruption and text corruption. The observations are as follows:

**(1)** The experimental results indicate that current MLLMs, when subjected to image and text corruption, do not exhibit significant decreases in accuracy in absolute terms. However, it is the portion

Table 10: **Results of robustness.** Acc represents the original accuracy without corruptions; Acc$_{crp}$ represents the accuracy after image and text corruptions; RRM% is Relative Robustness for multi-choice; Avg. is the weighted average results on ScienceQA and MMBench; As Kosmos-2† degenerates into random guessing, the results are meaningless. The entries that are both bold and underlined indicate the best performance.

| Scenario MLLM | ScienceQA | | | MMBench | | | Avg. | | |
|---|---|---|---|---|---|---|---|---|---|
| | Acc ↑ | Acc$_{crp}$ ↑ | RRM% ↑ | Acc ↑ | Acc$_{crp}$ ↑ | RRM% ↑ | Acc | Acc$_{crp}$ ↑ | RRM% ↑ |
| **LLaVA** | 46.55 | 39.12 | 30.88 | 44.13 | 33.16 | 33.76 | 45.66 | 36.92 | 31.94 |
| **LAMM** | 52.75 | 40.11 | 25.43 | 44.47 | 34.69 | **42.13** | 49.70 | 38.11 | 31.58 |
| **MiniGPT-4** | 47.00 | 38.37 | 22.95 | 54.34 | 35.88 | 31.04 | 49.70 | 37.45 | 25.93 |
| **mPLUG** | 48.44 | 40.70 | 38.77 | 49.57 | 29.85 | 10.36 | 48.86 | 36.70 | 28.31 |
| **Otter** | 50.22 | 39.07 | 22.68 | 53.91 | 35.29 | 29.31 | 51.58 | 37.68 | 25.12 |
| **LAv2** | **54.34** | **43.38** | **40.88** | 57.06 | 33.84 | 21.26 | 55.34 | 39.87 | 33.66 |
| **InstructBLIP** | 55.18 | 41.89 | 31.42 | **65.73** | **43.03** | 40.51 | **59.07** | **42.31** | **34.77** |
| **Shikra** | 45.21 | 36.59 | 8.40 | 63.26 | 38.35 | 30.20 | 51.86 | 37.24 | 16.43 |
| **Kosmos-2†** | 34.60 | 35.67 | 10.83 | 25.60 | 27.33 | 12.18 | 31.29 | 32.29 | 11.33 |

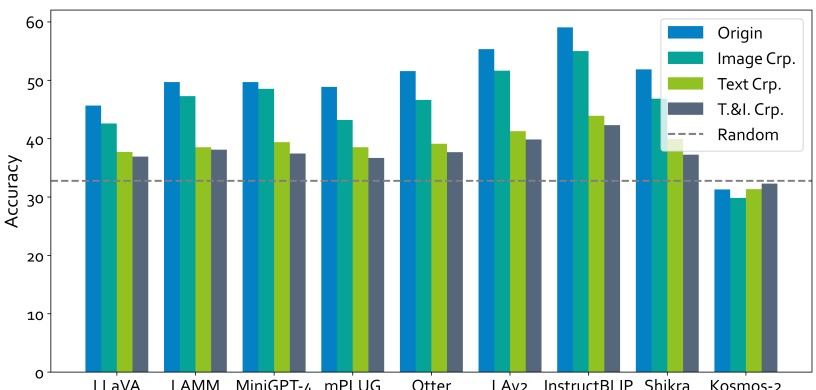

Figure 13: **Results of robustness under different settings.** The accuracy in this figure represents the weighted average results on ScienceQA and MMBench. The origin represents the original accuracy; Image Crp. represents the accuracy after image corruption; Text Crp. represents the accuracy after text corruption; I.&T. Crp. represents the accuracy after both image and text corruption; the dotted line represents the accuracy of random guessing.

of accuracy beyond random guessing that truly reflects the model's capabilities, and this portion is retained at less than half after perturbation. Given the prevalence of perturbations in daily environments, evaluating a model's robustness becomes pivotal.

**(1)** Image corruptions have a relatively minor effect on model performance, possibly owing to the robustness of the pre-trained vision encoder. In contrast, text corruptions show a significant impact on performance, potentially due to the heightened sensitivity of the MLLMs' text encoder when incorporating the visual tokens.

## E.6 HALLUCINATION

Table 11 presents the evaluation results for different difficulty levels of hallucination. Among them, LLaVA, LAMM, mPLUG-Owl, and Kosmos-2 exhibit more severe hallucination issues, as they tend to answer "Yes" very easily. This leads to nearly 100% Recall but with Acc and Precision both close to 50%, akin to random selection. Apart from these four models, the other models achieved relatively meaningful results. Overall, InstructBLIP achieved the best results, while Shikra also performed competitively, with an average accuracy being only 1.72% lower than InstructBLIP's.

Table 11: **Results of Hallucination.** Acc represents the accuracy of prediction; Precision represents how many of the predicted positive samples are true positive samples; Recall represents how many of all true positive samples are correctly identified; and Yes% represents the probability that the model outputs a yes answer. The entries that are both bold and underlined indicate the best performance.

| Dataset | MLLM | Acc | Precision | Recall | F1 Score | Yes% |
|---|---|---|---|---|---|---|
| MSCOCO-Random | LLaVA | 51.55 | 51.55 | 100 | 68.03 | 100 |
| | LAMM | 53.84 | 54.12 | 52.91 | 69.19 | 95.53 |
| | Minigpt4 | 80.93 | 89.67 | 71.20 | 79.38 | 40.92 |
| | mPLUG | 55.81 | 53.85 | 99.80 | 69.95 | 95.53 |
| | Otter | 82.27 | 89.11 | 74.73 | 81.29 | 43.23 |
| | LAv2 | 75.40 | 69.54 | 93 | 79.58 | 68.93 |
| | InstructBLIP | **90.24** | 93.55 | 87.06 | 90.19 | 47.97 |
| | Shikra | 87.18 | 87.00 | 88.33 | 87.66 | 52.33 |
| | Kosmos-2 | 51.55 | 51.55 | 100 | 68.03 | 100 |
| MSCOCO-Popular | LLaVA | 50 | 50 | 100 | 66.67 | 100 |
| | LAMM | 50 | 50 | 99.93 | 66.65 | 99.93 |
| | Minigpt4 | 74.3 | 75.4 | 72.13 | 73.73 | 47.83 |
| | mPLUG | 49.97 | 49.98 | 99.8 | 66.6 | 99.83 |
| | Otter | 73.57 | 73.03 | 74.73 | 73.87 | 51.17 |
| | LAv2 | 59.10 | 55.42 | 93 | 69.45 | 83.90 |
| | InstructBLIP | **83.37** | 81.07 | 87.07 | 83.96 | 53.7 |
| | Shikra | 83.3 | 80.25 | 88.33 | 84.10 | 55.03 |
| | Kosmos-2 | 50 | 50 | 100 | 66.67 | 100 |
| MSCOCO-Adverarial | LLaVA | 50 | 50 | 100 | 66.67 | 100 |
| | LAMM | 50.13 | 50.06 | 99.60 | 66.64 | 99.47 |
| | Minigpt4 | 72.17 | 72.51 | 71.40 | 71.95 | 49.23 |
| | mPLUG | 50.06 | 50.03 | 99.80 | 66.65 | 99.73 |
| | Otter | 70.07 | 68.35 | 74.73 | 71.40 | 54.67 |
| | LAv2 | 56.77 | 53.92 | 93 | 68.27 | 86.23 |
| | InstructBLIP | **80.63** | 77.14 | 87.07 | 81.80 | 56.43 |
| | Shikra | 79.27 | 74.78 | 88.33 | 81 | 59.07 |
| | Kosmos-2 | 50 | 50 | 100 | 66.67 | 100 |

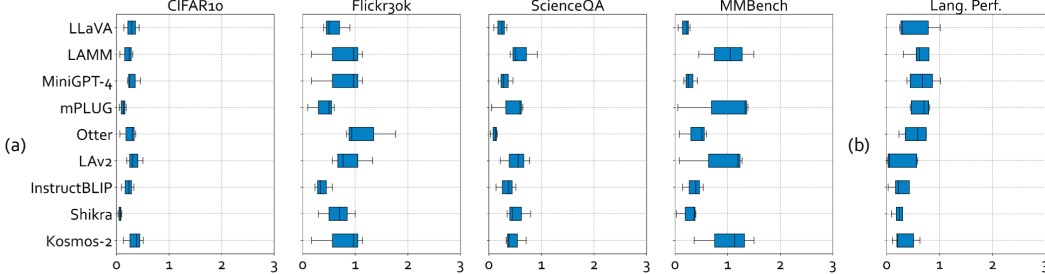

Figure 14: **Variance across seeds.** (a) Experiments are conducted on CIFAR10, Flickr30k, ScienceQA, and MMBench utilizing various random seeds. (b) Experiments of Language Performance. The results show the deviation from the mean score of 5 sampled evaluation evidence. The black line within each boxplot represents the median.

## F  CHEF PROVIDES RELIABLE ASSESSMENTS OF DESIDERATA

Due to the modular design of ChEF, we have the flexibility to employ different *Recipes* for evaluating the same *Scenario* and finally identify the most reasonable *Recipe* that can provide reliable and indicative assessments through experiments. Besides the reliability of evaluating the visual performance, we also try to ensure the stability and reliability of evaluating the desiderata. We conduct experiments to investigate the inherent randomness within them. This entailed scrutinizing the consistency of random factors, such as the utilization of random retrieved ICE for *Instruction* in ICL evaluation. Additionally, the evaluation of language performance, which is based on GPT assessment, inherently incorporates stochastic elements.

To evaluate the stability of random `ICE` as *Instruction*, we conduct experiments on CIFAR10, Flickr30k, ScienceQA, and MMBench, employing a diverse set of random seeds. To emphasize deviations from the mean value, we first calculate the average of results from the five different seed sets. Then, for each seed, we determine the deviation by subtracting this average and taking the absolute value of the difference. This approach highlights the variation in results for each seed compared to the average. As illustrated in Figure 14(a), the deviation for most model results is at around 1.0, indicating notable stability.

To mitigate systematic errors in GPT evaluation, we employed Multiple Evidence Calibration. In this approach, we prompt GPT-4 to provide evaluation explanations as evidence for deriving the final score, as described in Section C.4. As illustrated in Figure 14(b), our prompts effectively ensure the stability of GPT's scores across multiple samplings, where the maximum deviation is controlled under 1.0. This implies that GPT can maintain a consistent scale across multiple evaluations. Furthermore, the use of five sampled responses is deemed sufficient for GPT to furnish reliable and meaningful language performance scores.

These results indicate that the *Recipe* we provide for evaluating the desiderata is indicative and reliable.