# OpenReview forum: "ChEF: A Comprehensive Evaluation Framework for Standardized Assessment of Multimodal Large Language Models"
_ICLR.cc/2024/Conference — Submitted to ICLR 2024_

### Official Review · Reviewer_ZiWA · 2023-10-29

**Soundness:** 3 good
**Presentation:** 3 good
**Contribution:** 3 good
**Rating:** 6
**Confidence:** 4

**Summary:**

This paper proposes a comprehensibe evaluation framework ChEF for evaluating Multimodal Large Language Models.  ChEF consists of four modular components and allows for versatile evaluations in a standardized manner by designing new "recipes". The authors conduct evaluation of nine MLLMs  across various scenarios.

**Strengths:**

1. ChEF is modularly designed with four components, Scenario, Instruction, Inferencer, and Metric, which facilitates versatile evaluations in a standardized framework and easy set up pf new evaluations.
2. ChEF evaluates six capabilities that a competent MLLM model should possess, through constructing corresponding evaluation pipelines from a ChEF Recipe. These capabilities have not been systematically evaluated in exisiting MLLM Benchmarks.
3. The authors evaluate the generalizability of nine MLLMs across various scenarios and their composite capability for multimodal
interactions, and summarize valuable observations.

**Weaknesses:**

1. I am not certain if it is fair to incorporate current MLLM benchmarks into ChEF. These benchmarks have taken a significant amount of time to develop, so what is the core contribution of ChEF?
2. Besides in-context learning, ChEF only evaluates single-image input. However, the comprehension of multi-image input is also an important assessment dimension for MLLMs.

**Questions:**

See Weaknesses.

**Details Of Ethics Concerns:**

In my opinion, no ethics review are needed.

---

> ### Author Response · Authors · 2023-11-14
> **Response to Reviewer ZiWA**
>
> Before addressing your specific comments and questions, we would like to kindly inform you that we have provided an overall response to all reviewers. We believe that reading this response first will offer a comprehensive view of the revisions and clarifications we have made in light of the feedback received. We appreciate your time and effort in reviewing our paper. Following this note, I will proceed to address your specific concerns in detail.
>
> ### Q1: What is the core contribution of ChEF?
>
> We sum up the core contribution of our work in section 4 of our overall response. Various evaluation pipelines and criteria make fair comparison within the same evaluation pipeline a considerable challenge for MLLMs. ChEF effectively addresses these issues, enabling unified evaluation, and providing many interesting observations provided in our paper and supplementary materials. ChEF also propose evaluation on several capabilities that beyond visual performance, which have not been evaluated before.
> Additionally, ChEF is scalable, enabling users to extend their recipes for each component and supporting various evaluation pipelines of different dimensions of capabilities. These contributions assist users in assessing MLLM performance and guiding the improvement of capabilities in MLLMs.
>
>
>
> ### Q2: The comprehension of multi-image input is also an important assessment dimension for MLLMs.
>
> We are very grateful for your suggestion of this important capability dimension. However, it's important to understand that currently, very few open-source models support multiple image inputs, especially for multi-image VQA tasks, which require the insertion of image tokens into text tokens. This feature is not supported by most open-source models. Nevertheless, in ChEF, we rapidly implemented a multi-image evaluation recipe and conducted evaluations on the Winoground[1] dataset for four models that support multi-image input. Winoground is a dataset that proposes a task of correctly matching two given images and captions, which can be used to evaluate MLLMs' understanding of text-to-image references. The results are shown below:
>
> |**Setting**| **Model** | **Text** | **Image** | **Group** |
> | :-------: | :-------: | :-------: | :-------: | :-------: |
> |MLLM multi-image |**mPLUG-owl**|	36.25|	38|	31.25|
> |MLLM multi-image |**MiniGPT-4**|	26|	36	|22.75|
> |MLLM multi-image |**Kosmos-2**|	25.5|	32.5|	20.75|
> |MLLM multi-image |**Otter**|	22.75|	33.25|	18.75|
> |MLLM single-image|**PALI**| 	46.5|	38|	28.75|
> |MLLM single-image|**Blip-2** |	44|	26|	23.5|
> |CLIP-based       |	**VQ2**|	47|	42.2|	30.5|
> |	              / |**MTurk Human**| 	89.5|	88.5|	85.5|
> |	              / |**Random Chance**|	25|	25|	16.67|
>
>
> Winoground-Text/Image/Group are text, image and group score metrics proposed in Winoground. We prompt the MLLM with "Does image a match with the text b?" and calculate the probability that MLLM output "yes". Different from previous works where similarity was computed for individual image-text pairs, such as VQ2[2], PALI[3], and Blip-2[4]. In our experiment, we take two images and two captions together with a question as input to the MLLM models. Therefore, comparing the results directly with previous works may not be entirely fair due to the differences in experimental setup.
>
> The results indicates that the current MLLM models are not ideal in handling multiple images. It's noticeable that Otter, which was specially trained on ICL data, exhibits lower performance than the other MLLMs. This might be due to a significant gap between treating each image as an individual case and linking multiple images together in a task. There also exists a certain 'tug of war' in multi-image tasks.
>
> We are very grateful for your suggestion, which brings the interesting findings. We will incorporate more essential evaluation pipelines in the future. We also hope the community will provide more recommendations.
>
> [1] Winoground: Probing vision and language models for visio-linguistic compositionality.
>
> [2] What you see is what you read? improving text-image alignment evaluation.
>
> [3] Pali: A jointly-scaled multilingual language-image model.
>
> [4] Bootstrapping language-image pretraining with frozen image encoders and large language models.

---

> > ### Author Response · Authors · 2023-11-22
> >
> > We hope this message finds you well. We have noted the deadline for open discussion of ICLR 2024 is approaching, yet we have not yet received any feedback from you. In light of this, we sincerely wish to know if we can receive any updated comments regarding to our submission 177, titled "ChEF: A Comprehensive Evaluation Framework for Standardized Assessment of Multimodal Large Language Models". We are very pleased to hear from you on the reviewer's comments.

---

> > ### Comment · Reviewer_ZiWA · 2023-11-23
> >
> > I disagree with the claim that "very few open-source models support multiple image inputs" since one can simply flatten all image tokens of multiple images and append them before text tokens. Considerting that ChEF is a comprehensive evaluation benchmark, I tend to accept it and keep my rating.

---

### Official Review · Reviewer_n4Xu · 2023-10-31

**Soundness:** 3 good
**Presentation:** 3 good
**Contribution:** 2 fair
**Rating:** 6
**Confidence:** 5

**Summary:**

The authors propose a comprehensive assessment framework for large multimodal models using  four modular components and six "recipes" that stem from desiderata. They then apply the proposed framework to several state of the art large models and present many interesting insights on their performance.

**Strengths:**

1. The authors demonstrate a good understanding of the problem and lay out a comprehensive framework.
2. The overall proposed framework is wide ranging and thus leads to interesting insights.
3. The authors have been thorough in their implementation and experiments.

**Weaknesses:**

1. The paper does not justify its choices in a principled manner. The overall framework has an ad-hoc feel to it. While the reference are comprehensive, there is not enough logic to back up why those six desiderata for example are chosen and why some others are not. The work comes across as an engineering requirements style work rather than a scientific paper. I am open to being convinced otherwise. The field is moving very fast so just seemingly brute force evaluation of a bunch of models is not going to be helpful.
2. The writing needs to tone down the claims to being pioneering etc. Or at least back up such claims.

**Questions:**

1. What are the insights that drive your work? Please see the comments above on weaknesses.

---

> ### Author Response · Authors · 2023-11-14
> **Response to Reviewer n4Xu**
>
> Before addressing your specific comments and questions, we would like to kindly inform you that we have provided an overall response to all reviewers. We believe that reading this response first will offer a comprehensive view of the revisions and clarifications we have made in light of the feedback received. We appreciate your time and effort in reviewing our paper. Following this note, I will proceed to address your specific concerns in detail.
>
>
> ### Q1: There is not enough logic to back up why those six desiderata for example are chosen and why some others are not.
>
> We provide the reason for our choice of these desiderata in section 3 of our overall response. More details of these desiderata are supported in Supplementary Section C. **The principles of the six dimensions we selected are based on a survey and statistical analysis of the original LLM field, as well as the application of MLLM as an AI agent.**
>
> ### Q2: The field is moving very fast so just seemingly brute force evaluation of a bunch of models is not going to be helpful
>
> We provide the significance of ChEF and its contributions to the development of open-source MLLMs in section 2 and 4 of our overall response. We also agree that this research field is developing rapidly, especially with the strong capabilities demonstrated by API-only MLLMs like GPT-4V and Bard. We currently sampled some data and evalauted GPT-4V and Bard across multiple dimensions. Considering we couldn't access the probability outputs of these two models, and using GPT4 to evaluate GPT-4V's response for language performance evaluation seams unreasonable, we only evaluated them in terms of ICL, instruction following, robustness, and hallucination. We compared their performances with some open-source models on the same dataset, as shown in the table below:
>
> | **MLLM** | **ScienceQA** | **MMBench** | **ICL** | **Ins. Follow.** | **Robustness** | **Hallucination** |
> | :------: | :-----------: | :---------: | :-----: | :--------------: | :------------: | :---------------: |
> |**GPT-4V**|   **96.67**   | **93.80**   |  43.98* | **97.69**        | **82.16**      | **96.00**         |
> | **Bard** |  90.00        | 71.43       | 39.61*   |     71.41        | 71.05          |  88.88            |
> | **LLaVA**| 50.00         | 43.33       |**47.99**|       36.67      |  34.18         |  36.67            |
> | **Otter**|  63.33        | 50.00       | 47.91   |    44.44         |  37.35         | 80.00             |
> | **mPLUG-Owl**|  53.33    |  46.67      |   42.14 |     41.67        |  63.46         | 36.67             |
>
> As can be seen, GPT-4V nearly reaches the upper bound on several recipes proposed by ChEF, far exceeding the current open-source MLLMs. It's important to note that ICL is calculated as a relative metric of few-shot results compared to zero-shot results. For API-only MLLMs, their absolute performance in the few-shot setting is far superior to that of open-source models. In terms of VQA accuracy, instruction following, robustness, and hallucination, **there's a significant gap between open-source MLLMs and the two API-only MLLMs.**
>
> ChEF is mainly intended for the broad research community, aiming to inspire continuous improvement and advancement in open-source models. As an evaluation framework, **ChEF can guide open-source models in improving their performance in visual capabilities and trustworthiness, interactivity, and other competencies required by MLLMs.** Besides, ChEF supports a more interpretable implementation of MLLMs evaluation by modularizing each component. This is a contributor-friendly work with the goal of building a community where users can more easily build upon each other's contributions.
>
> ### Q3: The writing needs to tone down the claims to being pioneering etc. Or at least back up such claims.
>
> We claim that **ChEF is the first evaluation framework that covers existing benchmarks and provides expandable recipes for more traditional visual tasks, enabling unified evaluation and fair comparison of MLLMs across a variety of evaluation criteria.** Moreover, **we are the first to systematically evaluate MLLMs in dimensions beyond visual capabilities**, namely the six desiderata. However, we acknowledge that ChEF has limitations and is still in its preliminary stage. Despite its potential for expansion, the currently implementable recipes are limited and do not cover all possibilities. The six desiderata also do not encompass all other capability dimensions for evaluation, such as toxicity, privacy, societal, multilingualism, etc. Some methods of evaluation and datasets might not be the most appropriate. Nonetheless, we hope that our first attempt in building an evaluation framework and evaluation for the desiderata, will aid the development of the academic society. We also aim to expand our recipes in the future to include more evaluation pipelines, guiding the performance improvement of MLLMs.

---

> > ### Author Response · Authors · 2023-11-14
> >
> > ### Q4: What are the insights that drive your work?
> >
> > In the current development of MLLMs in the open-source community, we have identified that evaluating the capabilities of an MLLM is a challenging task, as claimed in section 2 of our overall response.
> > Therefore, **we aim to establish an evaluation framework that allows modules with different evaluation criteria to be compatible with each other.** This compatibility facilitates the establishment of various evaluation pipelines for **a unified assessment of models**, leading to many comparative findings. At the same time, we seek to evaluate metrics beyond visual capabilities, such as the trustworthiness and interactivity of MLLMs, to ensure that MLLMs **maintain the inherent capabilities required of a Large Language Model (LLM)**. Hence, we have designed new recipes for evaluating six dimensions of desiderata, leading to many interesting findings. We even discovered that visual capabilities are, to some extent, related to these desiderata, as mentioned in Section 3.5. We hope that our framework and the desiderata will guide the performance improvement of MLLMs, and these intensions drive my work.

---

> > > ### Author Response · Authors · 2023-11-22
> > >
> > > We hope this message finds you well. We have noted the deadline for open discussion of ICLR 2024 is approaching, yet we have not yet received any feedback from you. In light of this, we sincerely wish to know if we can receive any updated comments regarding to our submission 177, titled "ChEF: A Comprehensive Evaluation Framework for Standardized Assessment of Multimodal Large Language Models". We are very pleased to hear from you on the reviewer's comments.

---

### Official Review · Reviewer_mfBa · 2023-11-04

**Soundness:** 3 good
**Presentation:** 3 good
**Contribution:** 4 excellent
**Rating:** 8
**Confidence:** 4

**Summary:**

This paper introduces a comprehensive evaluation framework for Multimodal Large Language Models (MLLMs). The newly proposed framework has four modules, Scenario, Instruction, Inferencer, and Metric; and existing evaluation benchmarks can be summarized as recipes of the proposed framework. The authors conduct large-scale evaluations and presents valuable observations in the paper.

After rebuttal: I have read the rebuttal and I'd like to keep my scores.

**Strengths:**

This paper introduces a comprehensive evaluation framework for Multimodal Large Language Models (MLLMs). The newly proposed evaluation framework has a modular design, which allow it to recover various existing benchmarks with different recipes. Interesting observations are also presented in the paper.

**Weaknesses:**

Since the main contribution of this paper is introducing this new evaluation framework. I suggest the authors to add a section describing the system design/implementation of this framework in detail. It seems that such information is missing in the current draft.

**Questions:**

See above.

---

> ### Author Response · Authors · 2023-11-14
> **Response to Reviewer mfBa**
>
> Thank you for your insightful comments on the ChEF framework. I appreciate your perspective and would like to address your concern about the system design/implementation of ChEF.
>
> The criteria supported in each module, as well as their interrelationships, are shown in Figure 1(a) in our paper. We also provide the functions and details of each module's implementation in the Appendix Section B. To clearly present the entire evaluation pipeline, we provide details of the data flow, explaining how the data from scenario is transformed into input through instruction, and then how the inferencer enables the model to output answers for evaluation.
>
> ```
> 1. Load model and scenario. If ppl configured, each data sample in the scenario will include several predefined negative candidates. These negative candidates, together with the ground truth, form an `answer_pool`.
> 2. Configure the instruction, inferencer, and metric.
> 3. Iterate through each sample in the scenario. The instruction transforms each sample into an adaptive input for the model. When evaluating `sample_a`:
>     3.1 Get the input question from `sample_a`. For VQA tasks, a specific question is presented, while for other tasks, such as classification, there may be no question.
>     3.2 Concatenate the standard query or a specified query from the query pool with the question, along with the input image, to form `input_a`. If ICE is configured, retrieve a specified number of ICE from the scenario. Retrieval methods include random (random selection), fixed (selecting samples from the scenario via configured IDs), top-k images (based on similarity of images to the image of `sample_a`), and top-k text (based on similarity of text to the text of `sample_a`). The retrieval has following steps:
>       3.2.1 Retrieve a specified number of ICEs.
>       3.2.2 For each sample in ICE, concatenate the query with the question, to align with `input_a`.
>       3.2.3 Combine ICE and `input_a`, according to the MLLM's accepted format. For MLLMs that accept multiple images input, combine the entire ICE and `input_a`, while for MLLMs that accept only a single image, combine `input_a` with ICE without images, along with an additional system message.
>     3.3 If multi-turn inferencer is used, copy the input and replace the query in each with the corresponding multi-turn query.
> 4. Feed the inputs into MLLM and obtain the response, by outputing the probability token by token. There are several types of inferencer:
>     4.1 Direct inferencer: Choose the word with the highest probability for each token as the output.
>     4.2 CoT inferencer: Save the query in input as `original_query`, and replace that with a special query 'Let's think step by step'. Execute step 4.1, and then combine the response with `original_query` and execute 4.1 again.
>     4.3 PPL inferencer: Replicate the input, and append each candidate from the `answer_pool` to the query of each input. Feed to MLLM and calculate perplexity, and select the candidate corresponding to the input with the lowest perplexity as the response.
>     4.4 Multi-turn inferencer: Iterate the turn and exceute the corresponding inferencer.
> 5. Save the reponse results and execute the metric.
> ```
> Each component can be implemented to meet specific needs by passing a simple configuration. To more intuitively explain the design and implementation of our framework, we provide the pseudo code aligned with the data flow claimed above.
> ```
> # step 1
> model = get_model(model_name)
> scenario = build_scenario(scenario_name, ppl_cfg, **other_cfgs)
> # step 2
> instruction = build_instruction(instruction_type, prompt_cfg, icl_cfg, **other_cfgs)
> inferencer = build_inferencer(inferencer_type, **other_cfgs)
> metric = build_metric(scenario_name, **other_cfgs)
>
> answers = inferencer.inference(scenario, model, instruction) # step 3 and 4
>
> def inference(scenario, model, instruction):
>     answers = []
>     for sample in scenario:
>         input = instruction.generate(sample) # step 3
>         output = model.inference(input) # step 4
>         answers += output
>     return answers
>
> def generate(sample):
>     question = sample.get('question','') # step 3.1
>     # step 3.2
>     prompt = get_prompt(prompt_cfg)
>     input = prompt.format(question)
>     # step 3.2.1 to 3.2.3
>     ice = retriever.retrieve(sample)
>     ice = [(image, prompt.format(text)) for (image, text) in ice]
>     # input = [prompt_t.format(question) for prompt_t in multi_turn_prompt] # step 3.3
>     input = (ice, sample['image'], input)
>     return input
>
> def inference(input):
>     return model.direct_inference(input) # 4.1
>     # return model.CoT_inference(input) # 4.2
>     # return model.PPL_inference(input) # 4.3
>     # return model.multiturn_inference(input) # 4.4
>
> results = metric.metric(answers) # step 5
> ```
>
> We sincerely hope these provide sufficient clarity. We will also provide a more detailed tutorial and toolkit in the code release version, and include these details in the final version of our paper after revision.

---

> > ### Author Response · Authors · 2023-11-22
> >
> > We hope this message finds you well. We have noted the deadline for open discussion of ICLR 2024 is approaching, yet we have not yet received any feedback from you. In light of this, we sincerely wish to know if we can receive any updated comments regarding to our submission 177, titled "ChEF: A Comprehensive Evaluation Framework for Standardized Assessment of Multimodal Large Language Models". We are very pleased to hear from you on the reviewer's comments.

---

### Official Review · Reviewer_V16Y · 2023-11-06

**Soundness:** 2 fair
**Presentation:** 2 fair
**Contribution:** 3 good
**Rating:** 3
**Confidence:** 3

**Summary:**

This paper proposes ChEF, a framework for evaluating Multimodal Large Language Models (MLLMs). The main idea is to instantiate a “Recipe”, called “desiradata”, consisting of Scenarios (datasets), Instruction (how to pose questions such as in-context learning (ICE)), Inferencer (how an MLLM answers questions including Perplexity (PPL), Chain of Thought (CoT), and Multi-Turn), and Metrics.

They evaluate 9 MLLMs using 6 desiderata (9 Scenarios) that target measuring Calibration, In-context Learning, Instruction Following, Language Performance, Hallucination, and Robustness. See page 3 and Section 2.3 for more details.

**Strengths:**

- S1: The proposed ChEF framework is sound.

- S2: The experimental results are conducted on multiple models and settings and quite comprehensive.

**Weaknesses:**

- W1: Significance, Related work, and Execution. While I generally like the work that attempts to connect the dots and organize previous work, this work falls short. I do not think that the ChEF framework itself is a significant contribution as the 4 components of the Recipes are normally what people usually think about when it comes to evaluation. Thus, IMO, the main contributions lie in the instantiations of these Recipes or desiderata and their experimental results. However, the significance of this part is unclear due to two reasons.

  - W1.1: First, it is unclear both in the main text and in the supplement how this work is better than existing work in terms of “scalability” and “comprehensiveness” (cf. the first paragraph of the intro). The paper has to put more emphasis on the discussion of related work in order for the reader to understand the significance.

  - W1.2: Second, the desiderata in Section 2.3 themselves need more rationales/justification. Why do we care about these capabilities? Why do we instantiate them this way? For example, Hallucination consists of asking binary questions about the existence/absence of objects. Yet, this is not the only kind of hallucination. In general, it is unclear why the desiradata is what it is.

- W2: Clarity: related to W1, the paper would benefit from better presentation of desiradata. Perhaps having a table that lists down the 4 components. Justify why this is “versatile” evaluation.

**Questions:**

Please clarify as much as you can on my comments in Weaknesses.

---

> ### Author Response · Authors · 2023-11-14
> **Response to Reviewer V16Y**
>
> Before addressing your specific comments and questions, we would like to kindly inform you that we have provided an overall response to all reviewers. We believe that reading this response first will offer a comprehensive view of the revisions and clarifications we have made in light of the feedback received. We appreciate your time and effort in reviewing our paper. Following this note, I will proceed to address your specific concerns in detail.
>
> ### Q1: The main contributions lie in the instantiations of these Recipes or desiderata and their experimental results.
>
> Besides the recipes, desiderata, and the experimental results, the ChEF evaluation framework itself is also a significant contribution. Creating a benchmarking framework is an immensely challenging endeavor. We have elucidated this in Section 2 of our overall response. Indeed, after extensive research on numerous MLLM benchmarks, as well as conducting in-depth studies on works that explore instruction tuning and training methods, it became quite natural for us to decompose the evaluation pipeline into the four modules that we proposed. However, **no existing work has introduced this concept before, and none have successfully interlinked these modules in a compatible and cohesive manner.** ChEF achieves this features, thereby encompassing a wide range of existing benchmarks and enabling a unified assessment of MLLMs. We claim that the development of ChEF as an evaluation framework is a main contribution of our work. We are glad that Reviewer mfBa and n4Xu shares this perspective, recognizing the significant strides we have made in advancing the field.
>
> ### Q2: It is unclear both in the main text and in the supplement how this work is better than existing work in terms of scalability and comprehensiveness.
>
> We introduce several related works in Section 1 of our overall response, and more related works in Appendix A. We also have outlined these comparisons with other works in Figure 1 (b) of our paper. Compared to the previous works, ChEF firstly decomposed the evaluation pipeline into four components. We also provide a toolkit for a quick extension of each component.
>
> - **More scalable**: We provided an example to demonstrate the scalability. Reviewer Ziwa mentioned the need for multi-image understanding task evaluations in the comments. We chose Winoground[1] dataset and spent only several hours to complete the construction of the recipe and the evaluation of multiple MLLMs. We simply build the recipe by adding the dataset to the scenario, defining prompts for this task, and then utilizing PPL along with a metric for calculating accuracy from ppl results. The modular design allows us to build the new recipe easily. However, this process of expanding the recipe is difficult to implement in any current benchmark that provides a single undecoupled evaluation pipeline.
>
> - **More comprehensive**: ChEF has evaluated many traditional tasks and has also attempted to use more suitable methods for MLLM to evaluate detection tasks and fine-grained classification tasks, which have not been evaluated in any existing benchmarks. Additionally, ChEF has specially designed some unique recipes for evaluating the desiderata of MLLM, which have not been evaluated before. Based on these recipes, ChEF provides a comprehensive assessment of MLLMs.
>
> ### Q3: Why do we care about these capabilities? Why do we instantiate them this way?
>
> We provide the rationales and justification of these desiderata in Section 3 of our overall response. More details of these desiderata and the rationale for their evaluation are supported in Appendix Section C. **The principle of selecting these desiderata are based on a survey and statistical analysis of the original LLM field, as well as the application of MLLM as an AI agent.** We have specified the recipes for these desiderata by conducting survey on MLLM and related works in NLP. For example, for hallucination, we use the method proposed in POPE[2]. Other methods also reference some approaches from NLP, adapted for multimodal tasks. Of course, these recipes are just attempts to evaluate these desiderata. We welcome the community to expand the evaluation of desiderata by constructing new and more reasonable recipes through the expansion of each module of ChEF.
>
> ### Q4: Clarity
>
> The desiderata are evlauated through specially designed recipes. We illustrate the recipes in Figure 3 in our paper, and show distinguished design of each desideratum in Figure 4 in the paper. For a clearer and more intuitive understanding of the recipes, we list a table to show the four components of the recipe for each desiderata in Appendix Section D.3, Table 8. We sincerely hope these provides sufficient clarity.
>
> [1] Winoground: Probing vision and language models for visio-linguistic compositionality.
>
> [2] Evaluating Object Hallucination in Large Vision-Language Models.

---

> > ### Author Response · Authors · 2023-11-22
> >
> > We hope this message finds you well. We have noted the deadline for open discussion of ICLR 2024 is approaching, yet we have not yet received any feedback from you. In light of this, we sincerely wish to know if we can receive any updated comments regarding to our submission 177, titled "ChEF: A Comprehensive Evaluation Framework for Standardized Assessment of Multimodal Large Language Models". We are very pleased to hear from you on the reviewer's comments.

---

### Author Response · Authors · 2023-11-14
**Overall Response to all Reviewers**

We have identified some points of confusion or ambiguity in the paper and would like to clarify them here. We hope that all reviewers will first read this overall response before proceeding to their individual responses.

### 1. ChEF is a *Benchmarking Framework* rather than a *Benchmark*.

ChEF is distinct from existing benchmarks as it operates as a benchmarking framework. While some existing benchmarks [1, 2, 3] introduce well-designed evaluation datasets with specific evaluation pipelines tailored to their datasets, other works [4, 5, 6, 7] attempt to construct specialized evaluation pipelines for multiple visual tasks and various traditional datasets separately. However, the evaluation pipelines in these works cannot be decoupled.
In contrast, ChEF, serving as a benchmarking framework, decouples the evaluation pipeline into four indispensable modules: Scenario, Instruction, Inferencer, and Metric. Each module supports multiple different criteria. For example, in the Instruction module, the ICL section allows for the selection of in-context learning samples using different approaches. This design enables ChEF to incorporate existing benchmarking efforts as specific cases and facilitates the extension to select more suitable modules based on those benchmarks. The flexibility and scalability of this framework design empower ChEF to support fair evaluation and comparison of all MLLMs on existing benchmarks.

### 2. Building a benchmarking framework is significant.

**The lack of a unified evaluation framework has led to a stagnation in the development of the MLLM community.**

- **Lack of fair comparison**: Imagine the following scenario: as a researcher, I aim to propose a novel architecture design to enhance the performance of MLLMs on visual grounding tasks. To begin, I need to select a SOTA model to establish a strong baseline model. However, I observe that MLLMs' performance trends vary significantly across different benchmarks. For example, the InstructBLIP achieves an accuracy of 36%, ranking 9/13 on MMBench, while on SeedBench, it achieves an accuracy of 53.37%, ranking 1st out of 13. Similarly, MiniGPT-4 obtains a score of 1022.9, ranking 1/8 on LVLM-eHub, but performs poorly on all subtasks in MME. As a result, **based on the reported results of existing benchmark works, it is difficult for me to determine the relative performance of different MLLMs**. Consequently, it is challenging to select the strongest model as my baseline model.

- **Lack of scalability**: Next, I aim to define an evaluation pipeline that is better suited for assessing multiple visual tasks to reflect the effectiveness of my proposed method. MMBench provides a fine-grained categorization of visual abilities, covering 20 different ability dimensions, and I would like to leverage its data and task definitions. Additionally, although SEED-Bench has a narrower coverage in terms of data definitions, its utilization of model output probabilities is more reasonable. Therefore, I intend to employ SeedBench's inference method for the evaluation of MMBench's dataset. Furthermore, I plan to design instructions and metrics that are more suitable for evaluation. To the best of our knowledge, **expanding metrics or inference methods are not supported by any existing benchmark work**.

The lack of compatibility and integration among different works in the MLLM community hinders its progress and makes it challenging to initiate a snowball effect. We argue that the essential reason behind these issues is the absence of a unified evaluation framework capable of evaluating MLLMs with potential scenarios, instructions, inference pipelines, and metrics, while providing fair and equitable evaluation results.

---

> ### Author Response · Authors · 2023-11-14
>
> ### 3.Desiderata Design Principle
>
> **The principles of the six dimensions we selected are based on a survey and statistical analysis of the original LLM field, as well as the application of MLLM as an AI agent.** We may not have clearly explained this principle in our paper and the appendix. We will elaborate on it in detail next, and we will also include it in the final version of the paper after revision.
>
> Existing benchmark works lack a holistic capability evaluation for MLLMs. We usually liken the evaluation of visual tasks to specialized exams, while holistic capability is more akin to comprehensive abilities like morality and critical thinking. Specifically, we define and differentiate holistic capability as MLLM trustworthiness and interactive ability as an AI agent.
>
> - **Trustworthiness**: We conduct a comprehensive survey of the taxonomy and definitions of trustworthiness in the fields of AI governance and LLMs trustworthiness. We also count the number of times each dimension was referenced in the literature. We present the top six taxonomy dimensions based on the referred times in the table below.
>
>   Considering that evaluating fairness and safety in multimodal tasks requires extensive additional data collection and annotation, while interpretability requires exploring new techniques applicable to multimodal scenarios, we attempted to evaluate robustness and calibration. We also used the hallucination dimension to assess reliability. **Thus, considering their importance and their implementation in multimodal scenarios**, we have selected three desiderata for the current phase of MLLM trustworthiness evaluation, including calibration, robustness, and hallucination. Other capability dimensions that are difficult to evaluate at the current stage can be assessed in the future by establishing new recipes based on ChEF.
>
> - **Interactivity**: Considering the application of MLLMs as AI agents, we believe that an interactive MLLM should not only be able to perform computer vision tasks through question-answering but also **possess the ability to interact with humans**. It should understand human instructions (ICL), follow instructions accurately (instruction following), and maintain logical and easily understandable language expression during interactions (language performance). We argue that if an MLLM loses its ability to interact with humans and can only perform visual tasks, the significance of using MLLMs for multimodal research will be greatly diminished.
>
> The details of these desiderata and the rationale for their evaluation are provided in Appendix Section C. **These capabilities have not been evaluated in existing MLLM benchmarks before ChEF.** We hope that the design and observations of ChEF on these six desiderata can guide the future direction of the community's work in this area. We acknowledge that there are still many other desiderata, and as the performance of MLLM improve, more desiderata will need to be evaluated, such as toxicity, privacy, societal, multilingualism, etc. The evaluation of these capability dimensions can be easily achieved by expanding each submodule of ChEF to implement new recipes for evaluation.
>
> | Taxonomy  | Definition  | Reference | Referred Times |
> | --- | --- | --- | --- |
> | Robustness | Ability of a system to maintain its level of performance under a variety of circumstances. [8]  | 9,10,11,12,13,14,15,17  | 8 |
> | Fairness  |  Fostering diversity, AI systems should be accessible to all, regardless of any disability, and involve relevant stakeholders throughout their entire life circle. [15] | 9,10,11,12,13,15,16,17  | 8 |
> | Safety | AI systems need to be resilient and secure.[15] | 9,10,11,12,13,15,17 | 7 |
> | Calibration | Appropriate expression of model uncertainty.[14] | 9,10,12,14,16,17 | 6 |
> | Interpretability  | The ability to explain or to present in understandable terms to a human.[15] | 9,10,12,13,15  | 5 |
> | Reliability | Ability of an item to perform as required, without failure, for a given time interval, under given conditions. [8] | 9,10,11,12,17  | 5 |

---

> ### Author Response · Authors · 2023-11-14
>
> ### 4. Contribution
> We have reiterated our motivation and design principles in the previous section. Below, we will reiterate our main contributions.
> 1. We propose ChEF,  the first holistic framework designed to profile each MLLM and compare different MLLMs fairly. This framework addresses the previously noted gap of a standardized and comprehensive evaluation system for MLLMs. ChEF is scalable and comprehensive based on it's modular design, allowing for versatile and standardized evaluations, making it easier to integrate new evaluations by designing new recipes.
> 2. We intruduce six desiderata, focusing not only on specific task performance but also on broader aspects such as MLLM trustworthiness and interactive ability as AI agents. These desiderata are carefully selected based on their importance and their implementation in multimodal scenarios.
> 3. We proceed a comprehensive evaluation of nine prominent MLLMs across nine scenarios and six desiderata. This evaluation yields over 20 valuable observations about the generalizability and composite capabilities of MLLMs in various scenarios.
>
> [1] MMBench: Is Your Multi-modal Model an All-around Player?
>
> [2] MME: A Comprehensive Evaluation Benchmark for Multimodal Large Language Models.
>
> [3] SEED-Bench: Benchmarking Multimodal LLMs with Generative Comprehension.
>
> [4] LAMM: language-assisted multimodal instruction-tuning dataset, framework, and benchmark.
>
> [5] LVLM-eHub: A Comprehensive Evaluation Benchmark for Large Vision-Language Models.
>
> [6] VisIT-Bench: A Benchmark for Vision-Language Instruction Following Inspired by Real-World Use.
>
> [7] MM-Vet: Evaluating Large Multimodal Models for Integrated Capabilities.
>
> [8] iso/iec ts 5723:2022 Trustworthiness — Vocabulary.
>
> [9] AI Risk Management Framework.
>
> [10] A Taxonomy of Trustworthiness for Artificial Intelligence: Connecting Properties of Trustworthiness with Risk Management and the AI Lifecycle.
>
> [11] DecodingTrust: A Comprehensive Assessment of Trustworthiness in GPT Models.
>
> [12] Trustworthy LLMs: a Survey and Guideline for Evaluating Large Language Models' Alignment.
>
> [13] Cataloguing LLM Evaluations.
>
> [14] Holistic Evaluation of Language Models.
>
> [15] Ethics Guidelines for Trustworthy AI.
>
> [16] On Fairness and Calibration.
>
> [17] A Survey of Safety and Trustworthiness of Large Language Models through the Lens of Verification and Validation.

---

### Meta-Review · Area_Chair_EQj7 · 2023-12-05

**Metareview:**

The authors present a framework for benchmarking the performance of Multimodal Large Language Models (MLLMs) that is well thought out and modular, allowing authors to express how they tested their model according to the following criteria: Scenario, Instruction, Inferencer, and Metric.  There are two key contributions in this work, an evaluation framework and an exemplar or how that framework could be used to evaluate multiple key benchmark datasets.  Overall the reviewers do not disagree about the nature of the contribution but rather the significance of the contribution.  One reviewer is clear in their opinion that an evaluation framework alone is not enough of a contribution to ICLR.  Other reviewers are more positive, but even the reviewers with the highest rating feel that there is a lack of detail in the framework that would impede its widespread adoption,  specifically reviewer mfBa, (8, accept)  “I suggest the authors to add a section describing the system design/implementation of this framework in detail. It seems that such information is missing in the current draft.”

Although the authors give a very detailed response to the reviewers, most reviewers adhered to their original evaluation and did not think that this was enough of a contribution to meet the bar for ICLR in its current form.   The clarifications in the rebuttal represent a significant amount of work and we hope that incorporating these into a new submission will allow the paper to be accepted in another venue.  We would especially encourage the inclusion of the response to reviewer mfBa as this will hopefully make the path to adoption, and therefore impact, more clear.

**Justification For Why Not Higher Score:**

I believe that this framework is not very thorough.  It gives a lowest common denominator method of baselining different multimodal large language models, but I find the framework limited in its method of evaluation (restricted to multiple choice and binary yes/no answers). I did not find the evaluation of interaction convincing as it is simplistic and single turn.

**Justification For Why Not Lower Score:**

There is nothing harmful in this paper.  It contributes a framework that could be useful for benchmarking multimodal LLMs.

---

### Decision · Program_Chairs · 2024-01-16

Reject